# Surface passivation for highly active, selective, stable, and scalable $CO_2$ electroreduction

Jiexin Zhu [1,2,10], Jiantao Li [1,10], Ruihu Lu [3,10], Ruohan Yu[1,10], Shiyong Zhao[4], Chengbo Li[5], Lei Lv[1], Lixue Xia[6], Xingbao Chen[1], Wenwei Cai[1], Jiashen Meng[1,7], Wei Zhang[1], Xuelei Pan [1], Xufeng Hong[7], Yuhang Dai[1,2], Yu Mao[3], Jiong Li [8], Liang Zhou [1,9], Guanjie He [2], Quanquan Pang[7], Yan Zhao [6], Chuan Xia [5] ✉, Ziyun Wang [3] ✉, Liming Dai [4] ✉ & Liqiang Mai [1,9] ✉

Electrochemical conversion of $CO_2$ to formic acid using Bismuth catalysts is one the most promising pathways for industrialization. However, it is still difficult to achieve high formic acid production at wide voltage intervals and industrial current densities because the Bi catalysts are often poisoned by oxygenated species. Herein, we report a $Bi_3S_2$ nanowire-ascorbic acid hybrid catalyst that simultaneously improves formic acid selectivity, activity, and stability at high applied voltages. Specifically, a more than 95% faraday efficiency was achieved for the formate formation over a wide potential range above 1.0 V and at ampere-level current densities. The observed excellent catalytic performance was attributable to a unique reconstruction mechanism to form more defective sites while the ascorbic acid layer further stabilized the defective sites by trapping the poisoning hydroxyl groups. When used in an all-solid-state reactor system, the newly developed catalyst achieved efficient production of pure formic acid over 120 hours at 50 mA cm$^{-2}$ (200 mA cell current).

$CO_2$ conversion to value-added chemicals receives increasing interest from the scientific and industrial communities[1–6], which, compared to the current industrial production of chemicals from petroleum raw materials, is a cleaner and more sustainable process to valuable products. Electrochemical reduction of $CO_2$ ($CO_2$RR) using electricity from renewable energy represents a promising route for realizing the industrial conversion from $CO_2$ to value-added chemicals[7–11]. Due to the low selectivity towards multicarbon products, monocarbon products (e.g., CO, formic acid (HCOOH)/formate (HCOO$^-$)) are low-hanging fruits to be picked[12–15]. As can be seen below, however, it is still a big challenge to achieve a high formic acid conversion from $CO_2$ at wide voltage intervals and industrial current densities because the widely used Bi catalysts are often poisoned by oxygenated species.

[1]State Key Laboratory of Advanced Technology for Materials Synthesis and Processing, Wuhan University of Technology, Wuhan 430070 Hubei, P. R. China. [2]Electrochemical Innovation Lab, Department of Chemical Engineering, University College London, London WC1E 7JE, UK. [3]School of Chemical Sciences, The University of Auckland, Auckland 1010, New Zealand. [4]Australian Carbon Materials Centre (A-CMC), School of Chemical Engineering, University of New South Wales, Sydney, NSW 2052, Australia. [5]School of Materials and Energy, University of Electronic Science and Technology of China, Chengdu, P. R. China. [6]International School of Materials Science and Engineering, Wuhan University of Technology, Wuhan 430070 Hubei, P. R. China. [7]Beijing Key Laboratory for Theory and Technology of Advanced Battery Materials, School of Materials Science and Engineering, Peking University, Beijing 100871, P. R. China. [8]Shanghai Synchrotron Radiation Facility, Shanghai Advanced Research Institute, Chinese Academy of Sciences, Shanghai 201210, P. R. China. [9]Hubei Longzhong Laboratory, Wuhan University of Technology (Xiangyang Demonstration Zone), Xiangyang 441000 Hubei, P. R. China. [10]These authors contributed equally: Jiexin Zhu, Jiantao Li, Ruihu Lu, Ruohan Yu. ✉e-mail: chuan.xia@uestc.edu.cn; ziyun.wang@auckland.ac.nz; l.dai@unsw.edu.au; mlq518@whut.edu.cn

The industrial application of $CO_2RR$ requires the development of electrocatalysts with a high selectivity and durability at an ultra-high current density[16,17]. Bi is the most promising $CO_2RR$ catalyst with a high $HCOO^-$ selectivity[18-25]. Compared to Sn and Pb, Bi-based catalysts have a more suitable binding capacity to *OCHO, the intermediate of $HCOO^-$ pathway[20,26]. Because of their relatively positive redox potentials, Bi-based compounds show the universal valence reduction behavior during $CO_2RR$, accompanied by morphological reconstruction[19,20,27]. The negative potential and drastic structural reconstruction result in the formation of highly active defect sites or edge sites on the intricate surface of the post-reconstructed catalysts[19,20,28,29]. The defect-rich Bi catalysts thus formed generally show a high selectivity towards the formate production. However, these coordinately unsaturated defect sites could coordinate with some oxygenated species from the electrolyte, poisoning these active sites, just like the CO poisoning of noble metal catalysts, to drastically reduce their product selectivity and operation stability. For the industrialization of $CO_2RR$, therefore, it is highly desirable to ensure that the catalyst can operate over a wide voltage range and at high current densities with a high selectivity and stability[3,30,31].

Since catalytic reactions occur on the catalyst surface, surface modification can significantly regulate the process of catalytic reactions[32-40]. Molecularly modified surface layers could not only modulate the hydrophilic nature of the catalyst surface[35], but also stabilize certain key intermediates, such as $*CO_2^{\bullet-}$ and *OCCO[33,37]. This implies that microenvironmental modulation can control the adsorption behavior of ions and molecules on the catalyst, regulating the $CO_2RR$ process. Moreover, it is crucial to manage the adsorption configuration of the molecular layer on catalysts, as it can obstruct the adsorption of reactants to active sites if the molecules bind to those sites. The molecular layer adheres to the catalyst surface through electrostatic forces, generating a confined reaction space that enhances the activation behavior of the reaction intermediates at the interface between the catalyst and the molecular layer, thus offering more reliable feasibility[33-35].

Herein, we rationally regulate the reconstruction process of Bi-based catalysts by introducing an oxyphilic ascorbic acid (vitamin C, VC) molecular passivation layer (BS/VC hybrid catalyst) to prevent highly active defect sites from the hydroxyl poison, achieving a high formate production at the industrial ampere-level current density. Specifically, the oxyphilic molecule isolates the hydroxyl and defective Bi sites, thus ensuring the high formate selectivity and sustainability. The surface modification with VC also regulates the structural reconstruction to produce more formate-favor active sites. In situ X-ray absorption fine structure (XAFS) and attenuated total reflection surface-enhanced infrared adsorption spectroscopy (ATR-SEIRAS) reveal that the hydroxyl is captured by VC and isolated from the defective Bi sites. Density functional theory (DFT) calculations indicate that the defective Bi sites have the optimal selectivity towards formate production. As a result, this strategy achieves a faradaic efficiency (FE) above 95% over a wide potential range of 1.0 V and at the industrial ampere-level current density. A solid-electrolyte reactor devised from the BS/VC hybrid catalyst demonstrates the industrial-scale steadily continuous production of pure formic acid over 100 h.

## Results

### DFT calculations

To understand the HCOOH selectivity on different defect Bi sites, we carried out first-principles calculations of the $CO_2RR$ process on Bi catalytic sites. Computationally, we devised four Bi catalytic sites within intact Bi and Bi-vacancy defect environments (Fig. 1a), namely, p-Bi, sv-Bi, dv-Bi, and tv-Bi sites. On these Bi sites, the $2e^-$-$CO_2RR$ steps towards formic acid were investigated and corresponding Gibbs free energy changes were shown in Fig. 1b and Supplementary Fig. 1. The step associated with the *OCHO formation on Bi sites is the

rate-determining step (RDS). We found that p-Bi sites displayed a high free energy increase of 1.31 eV for the first hydrogenation to *OCHO, posing an overpotential of 1.23 V. Comparably, the double-coordinated Bi sites induced by Bi vacancies show better *OCHO adsorption, and generate a low overpotential of -0.09 V, certifying high $CO_2RR$ activity via Bi-vacancy. The size of vacancy has little effect on the *OCHO adsorption. Furthermore, compared to the computed $CO_2RR$ performance reported in previous literatures (Fig. 1c, Supplementary references 3-33), vacancy engineering endows the Bi sites, including sv-Bi, pv-Bi and tv-Bi, top-level $CO_2RR$ catalytic activities, approaching the volcano plot peak. This clearly shows the importance of Bi vacancies on the $CO_2RR$ performance, demonstrating that the presence of many Bi vacancies benefits high-performance $CO_2RR$.

The defective Bi sites are often derived from the reduction of Bi-based compounds, including chemical reduction and electrochemical reduction[13,41-43]. Due to the high activity of defect sites, it is likely to bind with species in the air or in the electrolyte solution, resulting in inactivation. From our calculations, we found that the sv-Bi shows a low binding energy to $O_2$ and $OH^-$ (Fig. 1d), leading to a decrease in the Bi vacancy density and weak *OCHO adsorption, and hence the catalytic performance degradation (Fig. 1e). Therefore, an attractive option to achieve both the high catalytic activity and stability for Bi catalysts is to mitigate the oxidation process. Ascorbic acid (VC) is a common antioxidant and, from our calculation results in Fig. 1d, we found that VC, compared to p-Bi and sv-Bi, has strong binding energies with $O_2$ and $OH^-$ species. Thus, incorporating VC into Bi-based catalysts could preferentially bind with $O_2$ and $OH^-$ species to prevent the oxidation of the Bi defect sites for enhancing the catalytic stability and selectivity.

### Materials synthesis and characterization

To obtain defect-rich Bi catalysts, we prepared ultralong $Bi_2S_3$ (BS) nanowires (Fig. 2a and Supplementary Figs. 2-4)[44]. Compared to the sputtering Bi (Bi-SC) catalysts (Supplementary Figs. 5-7), the BS showed better formate selectivity in $CO_2RR$ (Fig. 2b and Supplementary Fig. 8). Around 80% FE could be achieved at a potential range of 0.6 V and the formate was the only liquid product (Supplementary Fig. 9), while the remaining product is mainly $H_2$ with little CO. As shown in Supplementary Fig. 10, the BS was reduced to metallic Bi after $CO_2RR$. Interestingly, we found that the BS nanowires were converted into Bi nanosheets, and abundant defect sites like holes or edges formed (Supplementary Figs. 11-12). Atomic Force Microscope (AFM) revealed that the thickness of the reconstructed Bi nanosheets is about 4.7 nm (Supplementary Fig. 13). Similarly, the Bi-SC also showed a structural transformation from nanoparticles to nanosheets (Supplementary Fig. 14a), indicating that the derived Bi tended to exist in nanosheets structure. However, the Bi nanosheets derived from the Bi-SC showed an inconspicuous defect structure (Supplementary Fig. 14b).

From the mass spectrum (MS) of BS at -1.2 V, the signal of mass fragments for $HS^+$ ($m/z = 33$) and $H_2S^+$ ($m/z = 34$) are clearly observed (Supplementary Fig. 15). Under an optical microscope, we can track the variation of the BS morphology during $CO_2RR$. The nanowires structure of BS was identified in open circuit potential (Supplementary Fig. 16a). By applying a bias of -1.2 V, many microbubbles were produced, followed by the formation of many irregular-shaped nanosheets upon removal of the bias (Supplementary Fig. 16b-d). These findings illustrate that the $H_2S$ gas generated in situ aggravated the complete reconstruction of the BS nanowires. We envision that the reduction and reconstruction of the BS nanowires involved the diffusion and deposition of Bi atoms, and that the release of $H_2S$ gas promoted the formation of defect sites (Fig. 2c). During the long-term test, however, the BS showed a gradually declining current density and formate selectivity (Supplementary Fig. 17). After exposing to air for 1 h, the derived Bi catalysts showed obviously decreased formate

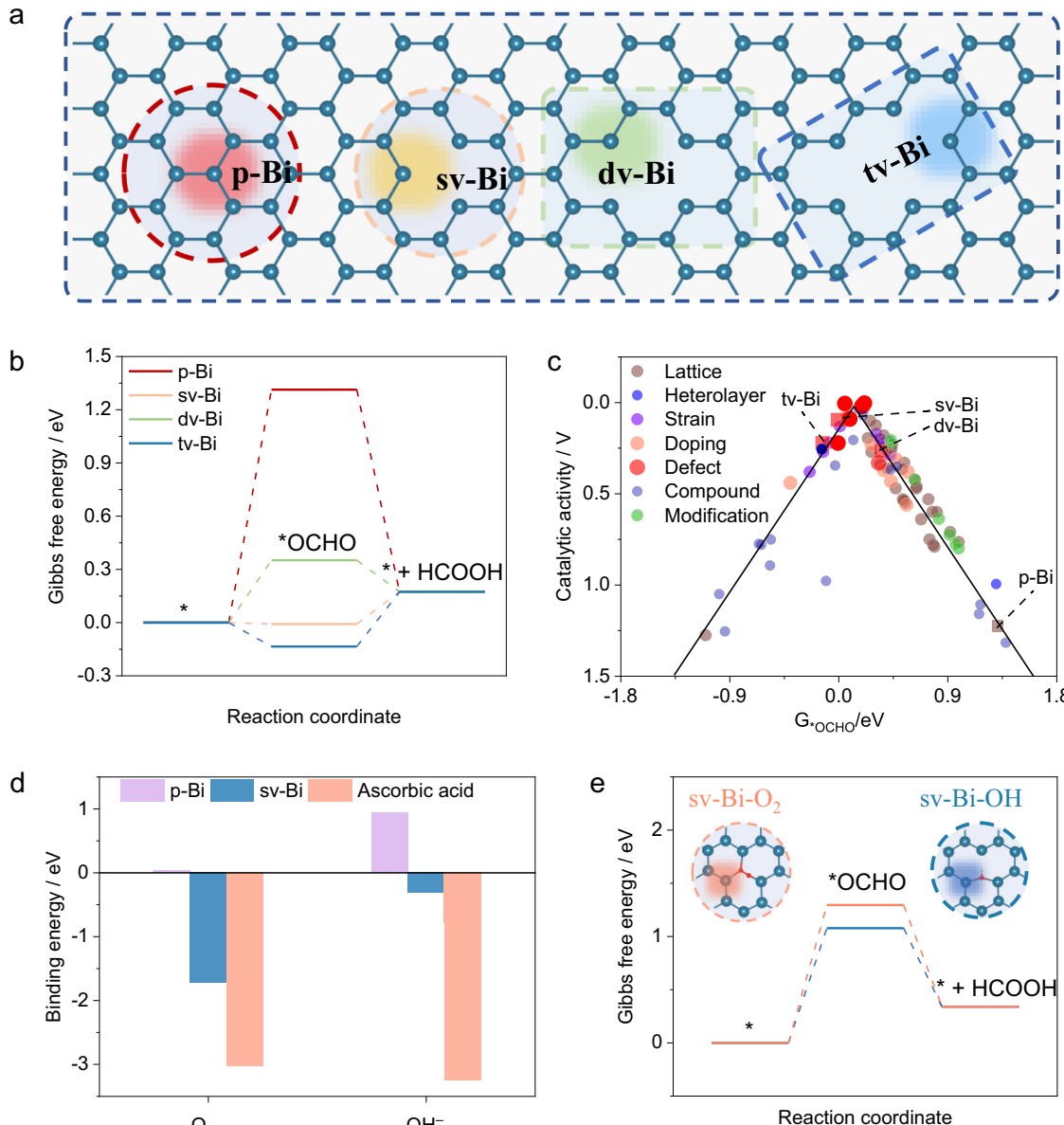

**Fig. 1 | DFT calculations. a** The Bi catalysts with different vacancy defects. The colored circles and highlights show the possible active sites. The Blue balls denote Bi atoms. **b** The Gibbs free energy profiles along $CO_2RR$ on the catalytic sites. **c** The volcano-shaped activity of $CO_2RR$ towards HCOOH on Bi-related catalysts. The theoretical catalytic performances of the Bi-related catalysts are from previous studies (Supplementary references 3–33). **d** The comparison of adsorbed $O_2$ and $OH^-$ species on the defect Bi catalytic sides and VC. **e** The Gibbs free energy profiles along $CO_2RR$ on the high defect Bi catalyst with adsorbed $O_2$ or $OH^-$ species. The white, red, and blue balls denote H, O, and Bi atoms, respectively.

selectivity and improved FE of $H_2$ (Supplementary Fig. 18). Clearly, therefore, the reduced formate selectivity was attributed to the oxidation of the defective Bi sites.

To improve the catalytic stability, we introduced an antioxidant adsorption layer of VC on the surface of the BS nanowires by electrostatic adsorption (Fig. 2d). With the VC coating, the BS/VC composite showed negligible morphology changes for the BS nanowires (Supplementary Fig. 19). A faint VC signal could be seen in the Fourier transform infrared (FT-IR) spectrum of BS/VC (Fig. 2e), indicating the successful introduction of VC, but the content is not high. High-angle annular dark-field scanning transmission electron microscopy (HAADF-STEM) image reveals a homogeneous molecular adsorption layer of 5 nm on the surface of the BS nanowires (Fig. 2f). Through adjusting the VC solution concentration, we found that the VC layer shows not much change in its thickness (Supplementary Fig. 20). As

shown in the STEM-electron energy-loss spectroscopy (STEM-EELS) images (Fig. 2g, h), the Z-contrast of Bi atoms shows an obvious bulk signal. In contrast, the C K-edge signal is mainly excited from the periphery of nanowires, indicating a tight VC coating. The C K-edge signals at the periphery (position 1) and center (position 2) display the same onset peak, but the signal at the center is lower (Fig. 2i), confirming that the VC was coated on the surface of BS nanowires. X-ray photoelectron spectroscopy (XPS) demonstrates that the intensity of the C−OH bond increased after the VC coating because VC contains abundant C−OH bonds (Fig. 2j, Supplementary Fig. 21), which is distinguished from possible adsorbates from the air. It is worth noting that there is no Bi−O bond observed. Considering that the thickness of VC is not affected by the concentration of VC solution, we believe that VC is mainly present on the BS surface in the form of physical adsorption via electrostatic interaction.

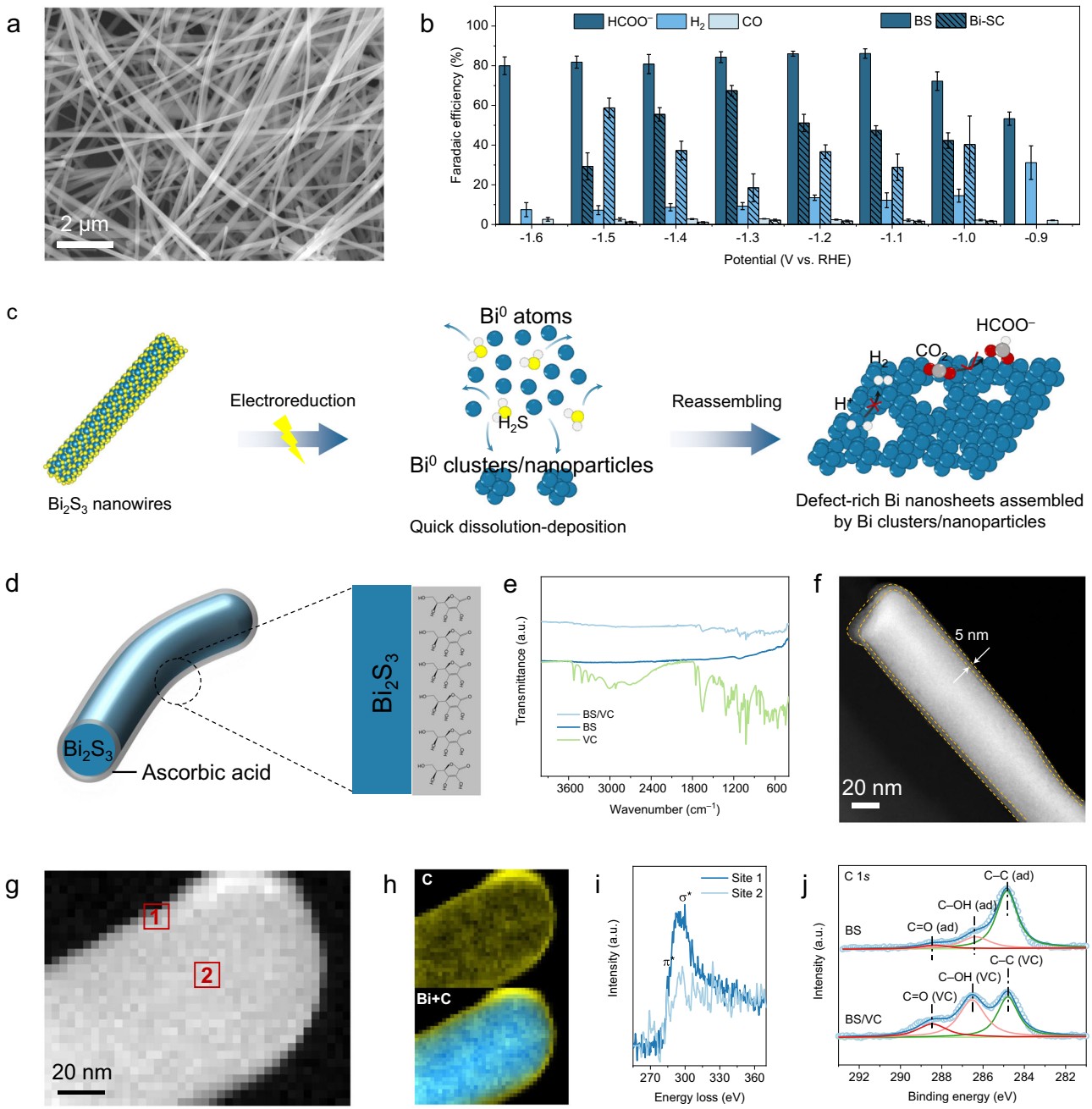

**Fig. 2 | Materials synthesis and characterization. a** scanning electron microscope (SEM) image of BS nanowires. **b** The FE of HCOO⁻, CO, and H₂ for BS nanowires and Bi-SC. The error bars represent the standard deviation of three measurements. **c** Schematic illustration of the electrochemical reconstruction of BS nanowires. **d** Scheme of cross-sectional structure of a BS/VC nanowire. **e** FT-IR analysis of BS/ VC, BS, and VC. **f** HAADF-STEM image of BS/VC. **g, h** HAADF-STEM image and corresponding C-K edge EELS mapping and Bi Z-contrast of BS/VC. **i** The C K-edge EELS for the red boxed area in (**g**). **j** High-resolution XPS C 1$s$ spectra for BS/ VC and BS.

## CO₂ electroreduction performance

The CO₂RR performance was evaluated in a flow cell electrolyzer with 1 M KOH electrolyte (Fig. 3a and Supplementary Fig. 22). The BS/VC showed an apparent increase in current density compared to BS in linear sweep voltammetry (LSV) curves (Fig. 3b). At the potential of −2.0 V vs. reversible hydrogen electrode (vs. RHE, unless otherwise specified), a current density up to 1.26 A cm⁻² was achieved. Like BS, only H₂ and CO were present in the gaseous product while HCOO⁻ is the only liquid product (Supplementary Fig. 23). On BS, the FE for H₂ in a flow cell is higher than that in the H-cell, and the FE for CO is almost the same (Fig. 3c). The FE for formate over most of the applied potential range is lower than 80% for BS (Fig. 3d). Considering the high

OH⁻ concentration in electrolyte, the descended formate selectivity can be attributed to the OH⁻ poison of the defective Bi sites. By contrast, it has a negligible effect on the BS/VC catalyst. The BS/VC exhibited a formate FE >90% at −0.7 V, which reached up to 98.6% at −1.1 V (Fig. 3d). A formate FE of 95% was achieved even when biased at −1.8 V. The FE of CO and H₂ are fairly low over all applied potentials for BS/VC. The high total current density and high FE enable an industrial ampere-level partial current density for formate production (Fig. 3e). A partial current density of 0.91 A cm⁻² for BS/VC was achieved at −1.8 V. At a high negative applied potential, the BS/VC shows a single pass CO₂ utilization efficiency of 4.5%, about twice of that for BS (Supplementary Fig. 24). Achieving the high product selectivity at a wide bias range and

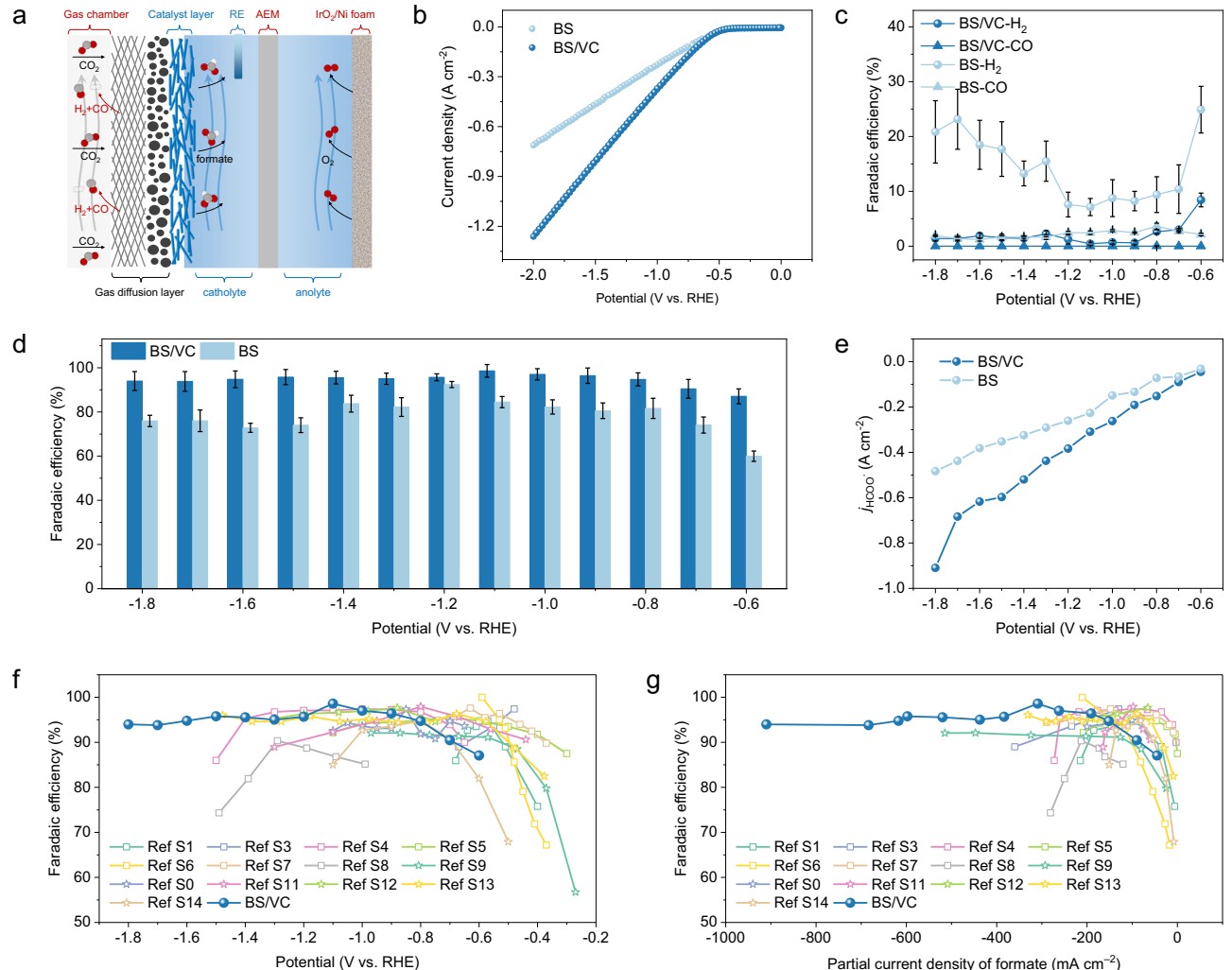

**Fig. 3 | CO2 electroreduction performance. a** Schematic representation of alkaline gas diffusion electrode system. **b** LSV curves of BS and BS/VC in 1 M KOH electrolyte. **c, d** The FE of $H_2$, CO (**c**), and $HCOO^-$ (**d**) for BS and BS/VC. The error bars represent the standard deviation of three measurements. **e** Partial current density of $HCOO^-$ under different applied potentials for BS and BS/VC. **f, g** Comparison of FE with working potential range (**f**) and partial current density of formate (**g**) with catalysts reported in the literature (the references are listed in Supplementary Table S1).

ampere-level current density is essential for industrial applications. The designed BS/VC catalysts exhibited over 95% formate FE at a potential range of 1.0 V (from −0.9 V to −1.8 V), which is wider than that for other reported catalysts (Fig. 3f and Supplementary Table 1). Also, the BS/VC outperformed the reported catalysts in term of the highest partial current density of formate (Fig. 3g and Supplementary Table 1).

We further investigated the antioxidation ability of BS/VC in air. During the $CO_2RR$ test, the $Bi^{3+}$ in BS/VC was also reduced to $Bi^0$ (Supplementary Fig. 10). Interestingly, after exposure to air for 1 h, the product selectivity for the derived Bi/VC showed negligible decay (Supplementary Fig. 18). We ascribed this increase in reactivity and product selectivity and air resistance to the protection of the VC layer to BS in the BS/VC catalyst, which blocks the adsorption of $OH^-$ and $O_2$ onto the defective Bi sites to impart the antioxidant effect.

**Mechanistic understanding**

To investigate detailed effects of the VC modification on the structural reconstruction of the BS nanowires and to gain the mechanistic understanding associated with the BS/VC catalysts, we conducted the ex situ TEM and in situ spectroscopic studies. The samples for ex situ TEM were collected by dropping the catalysts on Indium Tin Oxide (ITO) coated glass and proceed the $CO_2RR$ at 200 mA cm$^{-2}$ for 24 h in

$CO_2$-saturated 0.5 M $KHCO_3$. As shown in Fig. 4a and Supplementary Fig. 25a, c, the BS nanowires underwent a structure transformation from nanowires to nanosheets after the $CO_2RR$. In contrast, the BS/VC nanowires with a VC-modified layer retained the nanowire structure after electrolysis, but with a more pronounced pore structure and more defects of coordinately saturated Bi sites after the reconstruction (Fig. 4b and Supplementary Fig. 25b, d). As shown in Supplementary Fig. 26, the VC layer still existed on the surface of the derived Bi catalyst, though a less uniform VC coating layer after the $CO_2$ electrolysis (Fig. 4c).

To uncover the internal pore structure of the reconstructed Bi/VC nanowires, we implemented the three-dimensional (3D) visualization of tomographic reconstruction on a broken nanowire (Supplementary Fig. 27). The broken Bi/VC nanowire shows abundant pores on the surface, and the nanowire is seeming assembled by nanoparticles. From the front view, the Bi/VC nanowire just displays a porous structure (Fig. 4d) with a through-thickness hollow structure (Fig. 4e). The slice of 3D tomographic reconstructions clearly shows the solid structure apart from the through hole (Fig. 4f). The pore distribution analysis (Fig. 4g) reveals that the solid region of the Bi/VC nanowires is also porous structure to promote the diffusion of reactants and products. Without the restriction of a VC layer, the reduced $Bi^0$ atoms or

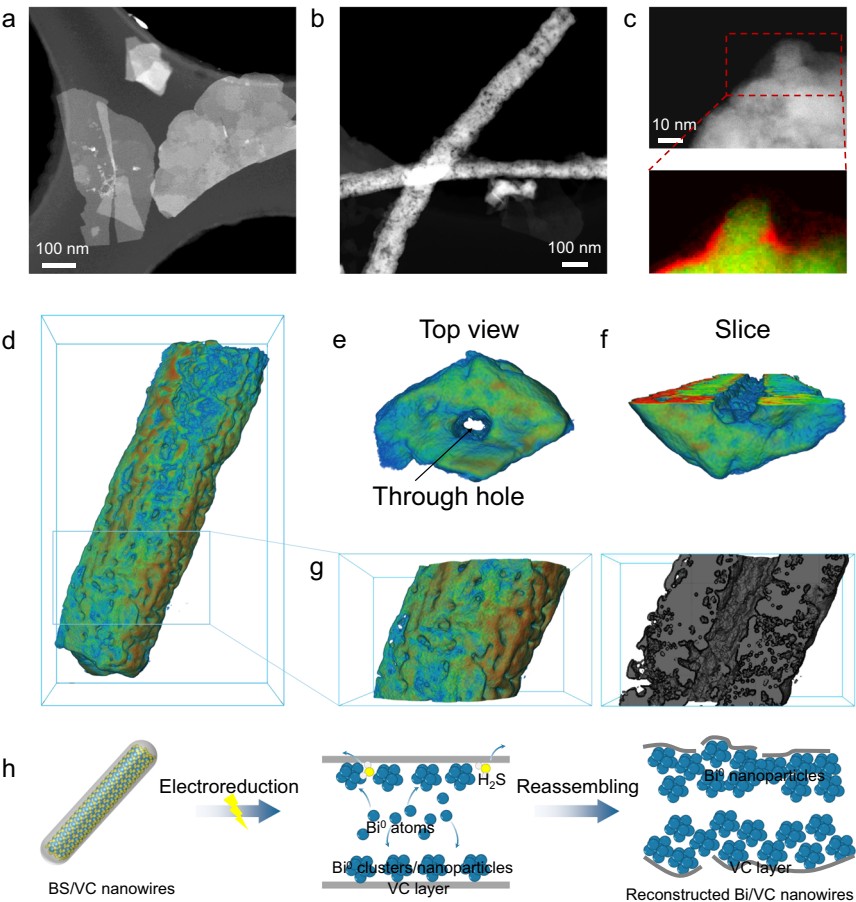

**Fig. 4 | Structural characterization of catalysts after CO₂RR. a** HAADF-STEM image of the BS-derived Bi nanosheets. **b** HAADF-STEM image of the BS/VC-derived Bi/VC nanowires. **c** HAADF-STEM image of the BS/VC-derived Bi/VC and corresponding C-K edge EELS mapping and Bi Z-contrast. **d**–**g** Three-dimensional visualization of tomographic reconstruction images of the BS/VC-derived Bi/VC. **h** Schematic illustration of the electrochemical reconstruction of the BS/VC nanowires.

nanoparticles would have packed freely to form the nanosheets. The VC layer protects/guides the assembling of Bi⁰ atoms/nanoparticles onto the inner surface of the VC layer to form a hollow structure through specific chemical interactions between $Bi_3S_2$ and VC to release $H_2S$ (Fig. 4h). Since the reorganization of Bi nanosheets is inhibited, a complete lattice cannot be obtained so that the resultant Bi/VC nanowires possess more defect sites to ensure a high current density and formate selectivity. An electrochemical active surface area (ECSA) test was conducted on catalysts before and after electrolysis to investigate the active area change during CO₂RR (Supplementary Figs. 28–29). The results show that with VC coating, the BS/VC exhibited a slight decline in active area, which could be attributed to the nanowire aggregation induced by the surface-attached molecules. After electrolysis, the reconstruction of nanowires led to an increase of active area, whereas the active area of BS/VC-after remained lower than that of the BS-after. The similar increasing trend indicates that the increase in active area is not solely responsible for the enhanced activity and selectivity for the BS/VC.

To reveal the valence and coordination environments of Bi in the BS and BS/VC, we performed in situ X-ray adsorption spectroscopy (XAS). Using a carbon paper (CP) as the loading substrate, the fluorescence signals of Bi $L_3$-edge under CO₂RR working conditions were collected (Supplementary Fig. 30). The X-ray absorption near-edge structure (XANES) spectra of Bi $L_3$-edge reveal that the Bi in BS and BS/VC hold a valence of $Bi^{3+}$ analogous to $Bi_2O_3$ (Fig. 5a, b). At the potential of 0 V, BS and BS/VC show negligible variation in the valence of Bi. At −0.6 V, the BS keeps the $Bi^{3+}$ state, while BS/VC shows an edge between

$Bi^{3+}$ and $Bi^0$, demonstrating the partial reduction of $Bi^{3+}$. When the potential is negative to −1.4 V, the Bi units in the BS and BS/VC are all reduced to $Bi^0$. The observed difference in valence change is attributable to the reducibility of VC induced by some S vacancies on the surface of the BS/VC (Supplementary Fig. 31). The coordination structure was ascertained by Fourier transforms extended X-ray absorption fine structure (FT-EXAFS). As shown in Fig. 5c, d, the FT-EXAFS spectra display a distinct Bi–S coordination at around 2.1 Å for the BS and BS/VC at the open circuit potential (OCP) and 0 V[45,46]. The BS still preserved the Bi–S coordination at −0.6 V. For BS/VC, it showed a partial reduction of $Bi_2S_3$ phase at −0.6 V, thus accompanying the disturbance of coordination environments. At −1.4 V, the Bi units in the BS and BS/VC were totally converted into Bi, and the difference is that the BS-derived Bi exhibited a similar coordination structure to the Bi foil, while the BS/VC-derived Bi showed a significant coordination structure perturbation. Without the VC layer, the Bi atoms quickly reorganized into a complete lattice, so no perturbed coordination structure was detected. The BS/VC-derived Bi exhibited a less unusual Bi–Bi coordination due to large perturbations at more negative potentials, including the dynamic reconfiguration processes and diffusion of products. Results from the corresponding wavelet transform (Supplementary Fig. 32) show that the BS/VC contains only the Bi–Bi coordination at −1.4 V, suggesting that the perturbed signal came from the dynamically reconfigured Bi atoms and not from other coordination structures. It is noteworthy that the BS exhibited a distinct Bi–O coordination at −1.4 V, most probably originated from the bonding of the defective Bi sites and OH⁻. Due to the VC layer protection, no Bi–O

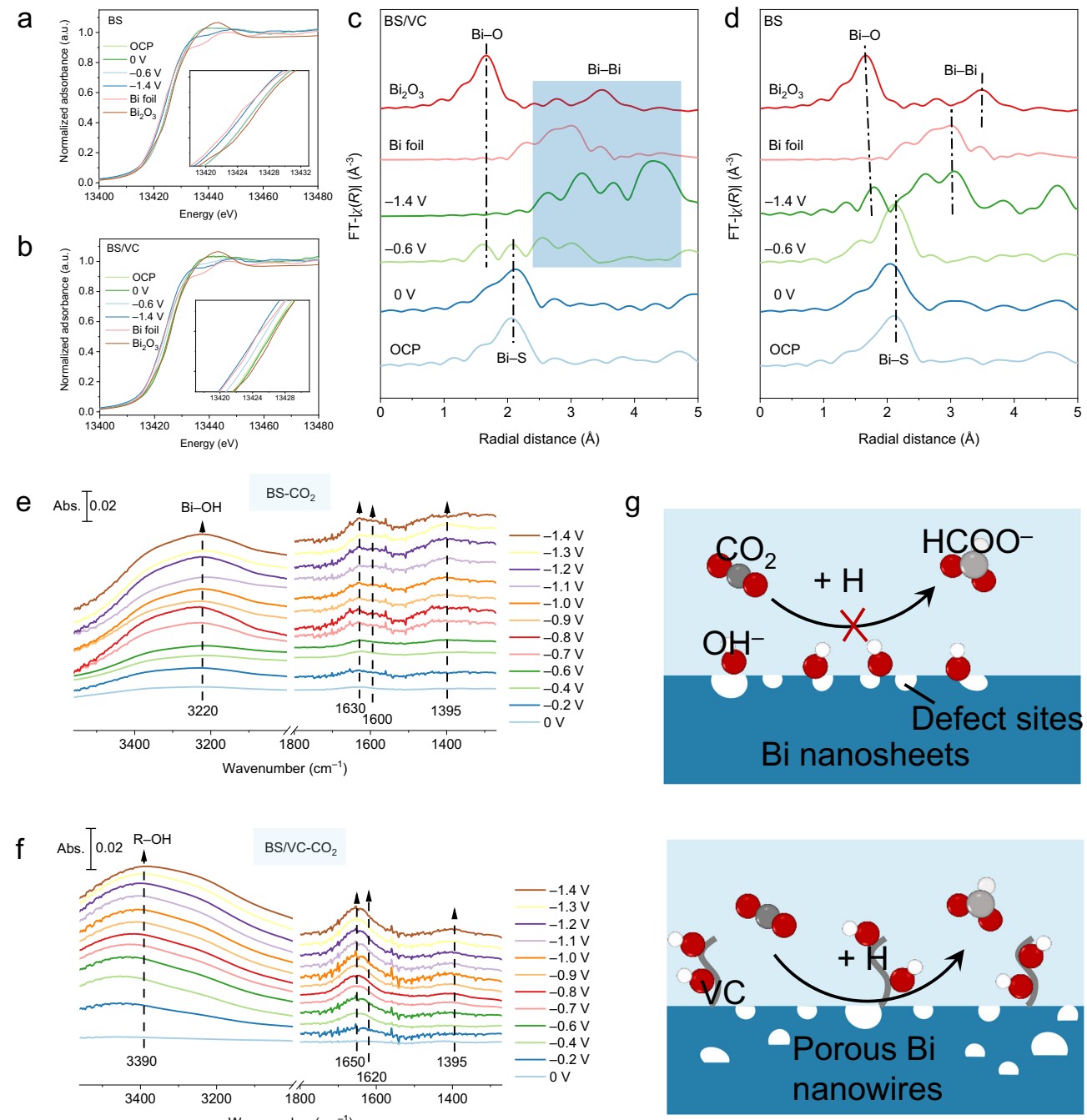

**Fig. 5 | In situ XAS and ATR- SEIRAS characterization. a**, **b** In situ Bi $L_3$-edge XANES spectra of BS (**a**) and BS/VC (**b**) under different applied potentials. Bulk Bi foil and $Bi_2O_3$ are listed as references. **c**, **d** In situ Bi $L_3$-edge EXAFS spectra of BS/VC (**c**) and BS (**d**) under different applied potentials. **e**, **f** In situ ATR-SEIRAS spectra of

BS (**e**) and BS/VC (**f**) at the different applied potentials (reference to RHE). **g** Schematic illustration of the hydroxyl trapping by VC passivate layer and ensure a smooth $CO_2$ reduction at the defective sites.

coordination was found in the BS/VC. Once again, these results clearly indicate the poisoning inhibition of the defective sites in BS/VC by the VC coating.

In situ ATR-SEIRAS is the most suitable technique to track the surface coordination environment of electrodes[47]. We implemented the in situ ATR-SEIRAS for the BS and BS/VC in 0.5 M $CO_2$-filled $KHCO_3$ electrolyte at a potential range from 0 V ~ −1.4 V (Supplementary Fig. 33). As shown in Fig. 5e, f, the two upward enhancement bands at 1395 and around 1600–1620 cm$^{-1}$ are attributable to the symmetric and asymmetric stretching vibration of COO radicals with a difference between the two bands >200 cm$^{-1}$, indicating that the COO radicals are

bound to the metal sites through a monodentate ligand, i.e., M-OCO[48,49]. The M-OCO pathway can only yield formate product. So, both the BS and BS/VC have the potential for high formic acid selectivity.

Since $CO_2RR$ is a proton-consuming reaction, a large amount of OH$^-$ was generated during the reaction, showing absorption peaks at 1630–1650 and 3600–3200 cm$^{-1}$ in the IR spectra. The BS mainly exhibited a broad absorption peak around 3220 cm$^{-1}$ (Fig. 5e). This low-frequency O−H absorption peak is due to the significant adsorption of OH$^-$ on the BS-derived Bi and the formation of a large number of intermolecular hydrogen bonds between OH$_{ads}$. When VC is present on

the surface, the VC will block the binding of OH⁻ to the defective Bi site. The OH⁻ has fewer binding sites on the VC and can only form fewer intermolecular hydrogen bonds, and thus the BS/VC exhibited an O–H broad absorption peak at the high-frequency position of 3390 cm⁻¹ (Fig. 5f). This demonstrates that the VC layer is effective in protecting the defective Bi sites from OH⁻ poisoning, as schematically shown in Fig. 5g. The defective Bi sites in BS are too strongly bound to OH⁻ and therefore cannot sustain high activity and selectivity. With a VC coating layer, the BS nanowires are reconstructed into Bi nanowires with abundant defect sites both inside and outside. The free OH⁻ prefers to bind to the outer VC layer, reducing the possibility of binding to defective Bi sites. The highly active defective Bi sites in the BS/VC enabled the highly selective conversion of $CO_2$, achieving an over 95% formate selectivity over an ultra-wide voltage range and at industrial ampere-level current densities.

The Tafel slope also provides valuable information for investigation of reaction mechanism. As shown in Supplementary Fig. 34, BS has a Tafel slope of 261 mV dec⁻¹, close to 200 mV dec⁻¹, indicating that a high barrier for the adsorption of $CO_2$ to active sites[50–52]. It is consistent with the poison of hydroxyl to active sites and prevent the adsorption of $CO_2$ molecules. A Tafel slope of 125 mV dec⁻¹ for BS/VC is indicative of a high barrier for charge transfer to form $*CO_2^-$ with no limitation for $CO_2$ adsorption[52–54]. This further demonstrates the effect of VC on the trapping or isolation of hydroxyl groups.

To further examine the distinctiveness of VC, we employed four types of molecular species (**I**: 4-Nitrothiophenol; **II**: 5,6-Dimethylbenzimidazole; **III**: 9-Anthracenecarboxylic acid; **IV**: 1-Dodecanethiol) on the surface of BS (Supplementary Figs. 35–38). None of these four molecules have antioxidant capacity, but **I** and **IV** possess hydrophobic groups. Compared to VC coating, the introduction of molecules **I**–**IV** resulted in a reduction in both the current density and the selectivity of formate to even lower than that of BS. The decline in current density can be attributed to their poor conductivity. The formate selectivity of BS/**I** or BS/**IV** was noticeably higher than that of BS/**II** or BS/**III**, but slightly lower than that of BS. We believe this is because the hydrophobic nature of **I** and **IV**, which facilitated the adsorption of $CO_2$ onto the catalysts. These results further highlight the importance of antioxidant to Bi-based catalysts. We also investigated the feasibility of this strategy for other formate producing catalysts (Supplementary Figs. 39–40). For this purpose, In and Sn electrodes were synthesized by sputtering metal In and Sn target on the gas diffusion electrode and VC layer was sprayed on the In and Sn electrode. In previous reports, the surface hydroxyl was demonstrated to promote the activation of $CO_2$ in neutral or weakly acidic electrolytes[55]. But with VC coating, In/VC showed a lower current density but enhanced formate selectivity in alkaline electrolyte at a high negative applied potential, indicating that there should be an optimal hydroxyl coverage for $CO_2$ activation. The Sn/VC showed a similar current density with Sn, but lower formate selectivity at a high negative applied potential, implying that the surface hydroxyl is important for the $CO_2$ conversion on Sn catalysts and the VC layer can affect the distribution of hydroxyl.

### All-solid-state electrochemical $CO_2RR$

As well known, the catalyst stability is often inaccurately judged in an alkaline flow cell due to the flooding problem[56–59]. In contrast, the all-solid electrolyte cell can convert $CO_2$ to pure formic acid without going through a complex product separation and purification step, which is free from flooding and can be used to accurately evaluate catalyst stability[13,60,61]. Pure formic acid was collected by slowly rinsing the solid electrolyte region with deionized water (Fig. 6a). As expected, the BS/VC-based all-solid electrolyte cell showed a high formic acid FE of 90% at a current density of 50 mA cm⁻² and 87% formic acid selectivity even when the current density increased up to 150 mA cm⁻² (Fig. 6b). The energy efficiency of formic acid as a function of current density reveals that the BS/VC achieved an energy efficiency of 33%,

and 22% can be maintained at high current density (Supplementary Fig. 41). At 50 mA cm⁻² (200 mA cell current), the BS/VC-based all-solid-state reactor showed a stable operation over 120 h with the HCOOH selectivity maintained above 80% (Fig. 6c). Even at a current density of 100 mA cm⁻² (400 mA cell current), the cell voltage and product selectivity remained stable for more than 80 h. However, when the current density increases to 200 mA cm⁻² (800 mA cell current), the formate selectivity shows rapid decline (Supplementary Fig. 42), which may be ascribed to the unstable adsorption of the VC layer at such a high current density. Clearly, therefore, the VC passivation layer in the BS/VC catalyst has effectively prevented the catalysts from the OH⁻ poisoning, and hence efficiently and consistently converted $CO_2$ to formic acid at the moderate industrial current.

## Discussion

We have developed a rational strategy to significantly enhance the $CO_2$ reduction durability of Bi-based catalysts by introducing an anti-oxidant passivation layer to eliminate the OH⁻ poisoning of the defective Bi sites. By modifying the surface of BS nanowires with oxygenophilic VC molecules, the BS nanowires were inhibited from the lattice reorganization to create a large number of defect sites with a monodentate ligand to bind the COO radicals, and hence the enhanced activity and selectivity. The outer VC layer prevented these highly active defective sites from hydroxyl poisoning, leading to a stable and highly selective conversion of $CO_2$ to formic acid over a long period (over 120 h). The OH⁻ trapping strategy enabled the BS/VC to achieve formate selectivity higher than 95% over a voltage range of more than 1 V at industrial ampere-level current densities. This work represents a breakthrough in developing advanced electrocatalysts for $CO_2RR$ and beyond, and highlights the importance of tailoring the catalyst microenvironment to significantly improve the catalyst performance with an enhanced activity, selectivity, and operation lifetime, thus promoting the industrialization of $CO_2$ conversion.

## Methods

### Chemicals

Bismuth nitrate pentahydrate ($Bi(NO_3)_3 \cdot 5H_2O$), Lithium nitrate ($LiNO_3$), Potassium nitrate ($KNO_3$), Sodium sulfide nonahydrate ($Na_2S \cdot 9H_2O$), hexadecyl trimethyl ammonium bromide (CTAB), thioacetamide (TAA), Potassium bicarbonate ($KHCO_3$), Potassium hydroxide (KOH) were purchased from Aladdin Industrial Inc. (Shanghai, China). Nafion solution (5 wt.%) was purchased from Alfa Aesar Chemical Co. Analytical grade ethanol was purchased from Sinopharm Chemical Reagent Co., Ltd. (Shanghai, China). All the chemicals were used without further purification.

### Materials synthesis

BS nanowires. The BS nanowires were synthesized according to the following procedure. 5 g $LiNO_3$, 10 g $KNO_3$, 0.485 g $Bi(NO_3)_3 \cdot 5H_2O$, and 0.48 g $Na_2S \cdot 9H_2O$ were put into the Teflon vessel and mixed uniform using a vortex mixer. 5 ml of deionized water was added to the Teflon vessel to regulate the viscosity of the molten salts. The vessel was sealed into the stainless-steel autoclave and subsequently annealed at 200 °C for 72 h. The obtained BS nanowire powder was washed several times with deionized water and ethanol before drying in a vacuum oven at 70 °C overnight.

BS/VC. BS/VC was synthesized according to the following procedure. 100 mg BS nanowires were mixed with 200 ml of ascorbic acid solution (0.5 M). After stirring for 24 h, the obtained BS/VC powder was washed several times with deionized water and ethanol before drying in a vacuum oven at 70 °C overnight.

Bi-SC. Bi-SC was synthesized by sputtering metal Bi target on the carbon paper (CP, TGP-H-060, Toray) using a magnetron sputtering machine (PD-200C, PDVACUUM Technologies Co., Ltd). The sputtering process was carried out at a power of 2 W for 5 min.

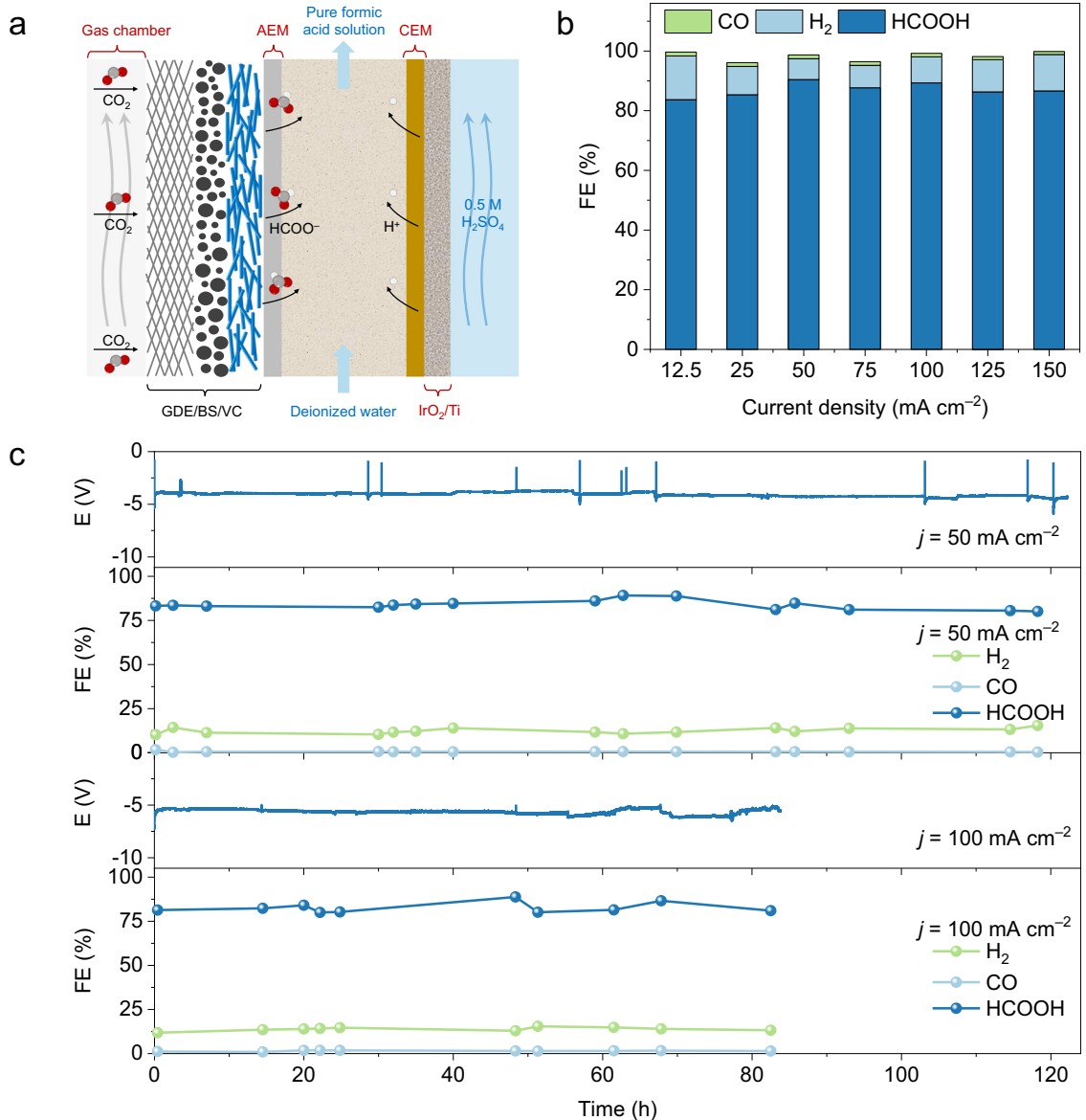

**Fig. 6 | All-solid-state CO2RR reactor performance. a** Schematic illustration of the all-solid-state electrochemical CO2RR reactor to produce pure formic acid. **b** FE of HCOOH, H₂, and CO for the BS/VC at different current densities. **c** Long-term production of pure formic acid from the BS/VC at current densities of 50 mA cm⁻² and 100 mA cm⁻², respectively, in the all-solid-state reactor.

## Characterization

The phase analysis of catalysts was performed by D8 Advance X-ray diffractometer using Cu-Kα radiation ($\lambda = 1.5418$ Å). JEOL-7100F scanning electron microscope (SEM) and double spherical aberration corrected transmission electron microscope (Titan Cubed Themis G2 300/Titan Cubed Themis G2 30) were adapted to perform the morphology, elemental distribution and three-dimensional visualization of tomographic reconstruction of catalysts. XPS measurements were performed on AXIS SUPRA, Kratos. Mass spectra was collected by GC-MS (Agilent 7890b/5977b).

## Electrochemical measurements

All electrochemical tests were conducted using an electrochemical workstation (Autolab PGSTAT 204, Metrohm). The CO2RR performance of the as-prepared catalysts was implemented in a H-cell, flow-cell, and an all-solid-state reactor. For the H-cell, the electrocatalyst powder inks were prepared using a mixture of 0.15 ml deionized water, 0.80 ml ethanol, 0.05 ml Nafion solution, and 5 mg of the catalysts,

followed by ultrasonication for 30 min. Then, 0.2 ml of the ink was uniformly loaded onto a CP with a catalyst loading of 1 mg cm⁻², which was used as the working electrode, while a Pt plate as the counter electrode and Ag/AgCl as the reference electrode. The working area of electrode was 1.0 cm². The working electrode and counter electrode were separated by a cation-exchange membrane (CEM, Nafion 117). 0.5 M KHCO₃ was used as electrolyte, and the flow rate of high-purity CO₂ (99.999%) was set to 20 ml min⁻¹ using a gas mass flow meter (CS200, Beijing Sevenstar flow Co., LTD). The potential values were referenced to the RHE according to the formula $E(\text{RHE}) = E(\text{Ag/AgCl}) + 0.198 + 0.059 \times \text{pH}$. All the potential was recorded without iR-correction. For the flow-cell, the electrocatalyst inks were prepared using a mixture of 20 mg catalysts, 1.9 ml ethanol, and 0.1 ml Nafion solution. The ink was sprayed onto the GDE to yield a catalyst loading of 1 mg cm⁻² (1.0 cm² electrode area). Ni foam and Ag/AgCl electrode were employed as the counter electrode and reference electrode, respectively. The working electrode and reference electrode were separated by an anionic exchange membrane (AEM, Sustainion®

X37-50 grade 60, Dioxide Materials). 1 M KOH was used as electrolyte, and the flow rate of $CO_2$ was set at 50 ml min$^{-1}$.

For the two-electrode cells with solid proton conductor for pure HCOOH solution production[60], an AEM (Dioxide Materials and Membranes International, Inc.) and a Nafion film (Fuel Cell Store) were used for anion and cation exchange, respectively. Around 0.5 mg cm$^{-2}$ catalysts loaded on YLS-30T GDL electrode (4.0 cm$^2$ electrode area) were used as a cathode, and $IrO_2$ was loaded on a titanium mesh as an anode. The cathode side was supplied with humified 30 s.c.c.m. of $CO_2$ gas (20 s.c.c.m. for stability test). The anode side was circulated with 0.5 M $H_2SO_4$ aqueous solution at 2 mL min$^{-1}$. The porous styrenedivinylbenzene sulfonated copolymer was used as the solid ion conductor[13]. DI water was used to release the produced HCOOH within the solid-state electrolyte layer, and the flow rate was kept at 1.4 mL min$^{-1}$ (0.6 mL min$^{-1}$ for stability test). All the measured potentials using a two-electrode setup were manually compensated.

## Product analysis

The gaseous products were detected by online GC (GC2014C, Shimadzu) equipped with a thermal conductivity detector (TCD) and flame ionization detector (FID) detector. Argon (Praxair 99.999%) was used as the carrier gas, and the GC was calibrated with $H_2$, CO, $CH_4$, and $C_2H_4$. Each gas phase product analysis time is set at 7 min. The FEs for gas products, such as $H_2$, and CO, were calculated as follows:

$$FE(\%) = \frac{NF \times \left(\frac{v}{60}\right) \times \left(\frac{y}{24.5 \times 10^9}\right)}{i} \times 100\% \qquad (1)$$

where $N$ is the number of electrons required for products ($N$ is equal to 2 for $H_2$, CO, and HCOO$^-$), y (ppm) is the volume concentration of the gas product, $v$ (sccm) is the gas flow rate (20 sccm for H-cell and 50 sccm for flow cell), $i$ (A) is the collected cell current, $F$ is the Faraday constant (96,500 C mol$^{-1}$).

Liquid products were quantified by analyzing the collected electrolytes using NMR (Bruker Avance III, 600 M). A known concentration of dimethyl sulfoxide (DMSO) was used as an interior label. Based on the $^1$H NMR analysis, the amounts of products were quantified by calculating the relative peak area of HCOO$^-$ (1.09 ppm) and DMSO (2.6 ppm). The FE for HCOO$^-$ was calculated as follows:

$$FE(\%) = \frac{nNF}{Q} \times 100\% \qquad (2)$$

where $n$ is the measured amount of HCOO$^-$ (mol), $N$ is the number of electrons required for each product, and $Q$ is the recorded total charge during the operation.

In solid-state reactor, the formic acid energy efficiency is calculated as follows:

$$EE_{full-cell} = \frac{\left(1.23 + \left(-E_{formic\,acid}\right)\right) \times FE_{formic\,acid}}{-E_{full-cell}} \qquad (3)$$

where $E_{full-cell}$ is the cell voltage applied in the solid-state reactor, and the $E_{formic\,acid} = -0.2\,V$[62].

## In situ X-ray absorption measurements

The Bi L$_3$-edge XANES and EXAFS spectra were collected at room temperature in fluorescence excitation mode at the beamline 12BM of Advanced Photon Source (APS) synchrotron radiation and beamline BL11B of the Shanghai Synchrotron Radiation Facility (SSRF). The electrocatalyst inks were prepared using a mixture of 0.1 ml deionized water, 0.85 ml ethanol, 0.05 ml Nafion solution, and 5 mg of the

catalysts, followed by ultrasonication for 30 min, and then 0.2 ml of the ink was uniformly loaded onto a CP. A commercial electrolytic cell (Zhongkewanheng) was used to accommodate the electrochemical reaction. $CO_2$-filled 0.5 M $KHCO_3$ was used as the electrolyte. Carbon rod and Ag/AgCl were used as the counter electrode and reference electrode, respectively. The electrochemical tests were conducted by electrochemical workstation (Autolab PGSTAT 204, Metrohm). The signal collection is achieved by performing the chronoamperometry test at the target potential for 10 min.

## In situ ATR-SEIRAS measurements

The in situ ATR-SEIRAS spectra were collected by an FT-IR spectrometer (Nicolet iS50, Thermo Scientific) equipped with an MCT-A detector. The catalyst inks were prepared using a mixture of 5 mg electrocatalysts, 0.95 ml ethanol, and 50 μl of Nafion solution. 10 μl of ink was dropped onto the central surface area of a hemicylindrical Si prism on which an Au film was chemically deposited. The Si prism was assembled in a spectroelectrochemical cell with a Pt wire as the counter electrode, Ag/AgCl electrode as the reference electrode, and 0.5 M $KHCO_3$ solution as the electrolyte. All spectrum was collected at a resolution of 4 cm$^{-1}$, and each single-beam spectrum was an average of 200 scans. An Autolab PGSTAT 204 electrochemical workstation (Metrohm) was used for potential control. High pure $CO_2$ is continuously introduced into the electrolyte during the reaction.

## DTF calculations

First-principle DFT calculations were conducted using Vienma ab initio simulation package (VASP)[63,64]. Using the electron exchange and correlation energy was treated within the generalized gradient approximation in the Perdew−Burke−Ernzerhof functional (GGA-PBE)[65], the calculations were done with a plane-wave basis set defined by a kinetic energy cutoff of 450 eV. The long-range dispersion interactions between adsorbates and surface was treated applying DFT-D3 method developed by grimme et al. [66]. The $k$-point sampling was obtained from the Monkhorst−Pack scheme with a (3 × 3 × 1) mesh for optimization. The geometry optimization and energy calculation are finished when the electronic self-consistent iteration and force were reach $10^{-5}$ eV and 0.02 eV Å$^{-1}$, respectively.

At standard condition ($T = 298.15$ K, pH = 0, and $U = 0$ V (vs. SHE)), the Gibbs free energy change $\Delta G$ is defined as the following equation:

$$\Delta G = \Delta E + \Delta E_{ZPE} - T\Delta S \qquad (4)$$

Where $\Delta E$ is the energy change obtained from DFT calculation, $\Delta E_{ZPE}$ is the difference between the adsorbed state and gas, which was calculated by summing vibrational frequency for all model based on the equation: $E_{ZPE} = 1/2\sum h\nu_i$ (T is the temperature (298.15 K) in the above reaction system, and $\triangle$S represents the difference on the entropies between the adsorbed state and gas phase. The entropies of free molecules were obtained from NIST database (https://janaf.nist.gov/).

The adsorption energy of *OCHO intermediates ($\Delta G_{*OCHO}$) was calculated based on the computational hydrogen electrode method, namely, the energy of H$^+$/e$^-$ pair is closely equal to half the energy of $H_2$ molecules[67]. $\Delta G_{*OCHO}$ was calculated according to following equations:

$$\Delta G_{*OCHO} = G_{*OCHO} - G_* - G_{CO2} - 1/2G_{H2} \qquad (5)$$

Where * represents the adsorbed sites associated with Bi catalysts. The above $\Delta G_{*OCHO}$ are defined as the reaction free energies of the following reactions.

$$* + CO_2 + H^+ + e^- \rightarrow *OCHO \qquad (6)$$

The two-dimensional Bi catalyst was modeled within 32 Bi atoms in a lattice constant of $18.18 \times 18.18$ Å$^2$. The high-defect Bi catalysts was represented by the combination of single (sv-Bi), double (dv-Bi), and triple (tv-Bi) Bi vacancies, which can be constructed by removing one, two, and three Bi atoms from pristine Bi catalysts, respectively. Meanwhile, we posited $O_2$ and $OH^-$ species on exposed Bi edge to simulate the effect of oxidation on $CO_2RR$. Furthermore, to consider the protection effect of ascorbic acid on avoiding the oxidation, we calculated the binding energy of $O_2$ and $OH^-$ on ascorbic acid referred to previous works[2] and then compare with that on Bi catalysts. we extracted the calculated $CO_2RR$ overpotential and $\Delta G_{*OCHO}$ values from previous works with regards to Bi-related catalysts (Supplementary reference 3–33) and plotted the volcano-shaped relation of catalytic activity against $\Delta G_{*OCHO}$.

## Data availability

All data supporting the findings of this study are available within the article and the Supplementary Information file. All raw data generated during the current study are available from the corresponding authors upon request.

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

## Acknowledgements

L.M. acknowledges the National Key Research and Development Program of China (2020YFA0715000), National Natural Science Foundation of China (51832004, 52127816), National Energy-Saving and Low-Carbon Materials Production and Application Demonstration Platform Program (TC220H06N). J.Z. acknowledges the Fundamental Research Funds for the Central Universities (195101005, 2020IIII004GX). L.D. acknowledges the Australian Research Council (FL 190100126, CE230100032). We thank the Beamline 12-BM-B at the Advanced Photon Source and Beamline BL11B at the Shanghai Synchrotron Radiation Facility for XAFS measurement. This S/TEM work was performed at the Nanostructure Research Center (NRC), which is supported by the State Key Laboratory of Advanced Technology for Materials Synthesis and Processing, and the State Key Laboratory of Silicate Materials for Architectures. The computational study is supported by the Marsden Fund Council from Government funding, managed by the Royal Society Te Apārangi. Z.W. and R.L. acknowledge the use of New Zealand eScience Infrastructure (NeSI) high performance computing facilities, consulting support and/or training services as part of this research. New Zealand's national facilities are provided by NeSI and funded jointly by NeSI's collaborator institutions and through the Ministry of Business, Innovation & Employment's Research Infrastructure programme. URL https://www.nesi.org.nz.

## Author contributions

C.X., Z.Y.W., L.D., and L.Q.M. conceptualized and supervised the project. J.X.Z. and J.T.L. conceived the idea and designed the experiments. J.X.Z., J.T.L., S.Y.Z., L.L., X.B.C., and W.W.C. conducted the synthesis, in/ex situ characterization, and flow-cell tests. R.H.L, L.X.X., Y.M., Y.Z., and Z.Y.W. carried out the density functional theory calculations. R.H.Y. conducted the TEM characterization. C.B.L. and C.X. performed the all-solid-state electrochemical test. J.X.Z., J.T.L., W.Z., X.L.P., and J.L. contributed to the in situ X-ray adsorption spectroscopy tests. J.X.Z., R.H.L, J.S.M., W.Z., X.L.P., X.F.H., Y.H.D., G.J.H., and Q.Q.P. discussed the experiment and density functional theory results. J.X.Z., J.S.M., L.Z., Q.Q.P., L.D., and L.Q.M. wrote and revised the manuscript. All authors discussed the results and assisted during manuscript preparation.

## Competing interests

The authors declare no competing interests.
