## [Peer Review File · Nature Communications]

REVIEWER COMMENTS

Reviewer #1 (Remarks to the Author):

The manuscript by Zhu et al. examines how chemical functionalization of the catalyst surface can be used to tailor the microenvironment near the active site. This study shows that vitamin C functionalization of a bismuth catalyst (wherein defect sites provide catalytically active sites) enhances the selectivity toward formic acid and extends the durability of the catalyst.

The authors provide a comprehensive characterization of the structure and surface chemistry of the defect-rich bismuth catalyst including, TEM, XPS, and FTIR.

The study is based on a close integration of experiment and computational simulation (i.e., binding energies from DFT).

Overall, the results are interesting and the mechanistic interpretation is, in large part, supported by the provided experimental and computational evidence. The authors convincingly show that oxidation of the defective Bi sites reduced the formate selectivity and that this effect can be mitigated by the functionalization with vitamin C (VC)

However, there are also several aspects that should be revised to improve the quality and impact of the study (i.e., accept after major revisions to address the points outlined below)

1. The authors point to several recent studies which showed that chemical modification of the microenvironment can impact adsorption behavior. While previous studies as well as the current study show an enhancement effect it seems that there should be an underlying trade-off. On the one hand functionalizing (i.e., passivating) the catalyst surface affords more control over chemical functionality (e.g., hydrophilicity to control adsorption of intermediates and spectator ions); on the other hand, if the functionalizing species is bound to the surface (active site), then this site is unavailable to activate the heterogeneous catalytic reaction. It would be helpful to discuss this trade-off and associated design principles to guide the optimal balance of functionality and activity.
2. The authors present a series of DFT calculations of binding energies on bismuth surfaces with varying defect densities. These calculations provide important initial insights into the role of defects which are subsequently examined in experimental studies. One of the most important aspect of these calculations is the strong binding of the ascorbic acid anti-oxidant to the catalyst surface. It would be helpful to provide more information on the binding configuration on the defect-rich bismuth surface. How and where is the VC bound? Figure 1d compares binding energies, but doesn't specify the defect type or where/how the VC is bound. Is VC chemically bound or adsorbed?
3. The effect of adding VC to maintain high formic acid selectivity is clearly evident from the presented results. However the catalyst design principles and impact of the paper could be significantly enhanced by further quantifying this effect. Beyond the A/B comparison of (A with VC , B without VC) it would be helpful to quantify this effect.

- a. What happens if the VC coverage is increased / decreased?
- b. What happens with other surface functionalizations? (i.e., species that also bind to the defect-rich Bi surface but don't have anti-oxidant effects?)

Addressing these questions would be very helpful and would also help to address the trade-off between functionality and activity described above.

4. Figure 3 shows remarkably high current densities as high as -1.2 A/cm^2 , yet the extended operating time measurements in Figure 6 show only 0.1 or 0.05 mA/cm^2 . How does the durability of the electrolyzer depend on the current density? Even if the durability at higher current density is reduced, this would still be an important and interesting result to share.
5. Figure 4 includes additional structure analysis including tomographic reconstruction of the nanowire. The data are nice, but not that critical to the main theme of the rest of the paper.
6. Figure 5g provides a nice schematic illustration of the mechanistic interpretation of the role of defect sites and how VC functionalization impacts the microenvironment near the active site. The bottom half of Figure 5g shows two different reaction environments; presumably the outside and inside of the hollow wire. I don't think the interpretation about the reaction on the hollow core of the wire is (a) necessary or (b) supported by the presented data, so I suggest to just focus on the outside surface.

Minor point:

- On pg 4, the authors state that "... catalytic reactions occur mainly on the catalyst surface, surface modification can significantly regulate the process of catalytic reactions." I would say that by definition heterogeneous catalytic reactions occur at the surface (active site), so the statement about 'mainly' should be removed, unless I misinterpreted the intended message.

Reviewer #2 (Remarks to the Author):

General remarks:

The authors report a Bi_3S_2 catalyst interface-modified with ascorbic acid for CO_2RR to formate. The manuscript introduces the hybrid catalyst delivering outstanding catalytic activity. The hybrid catalyst

showed very high FE at impressive ampere current densities with stability over 120hrs (in a solid-electrolyte reactor system). The presented system competes well with state-of-the-art Bi-based electrocatalysts reported.

The origin of the catalytic properties is then discussed on a mechanistic basis. The authors have used SEIRA spectroelectrochemistry to monitor the BiS-VC surface under potential. It is concluded that the surface functionalisation achieved by layering ascorbic acid is claimed to stabilise the BiS (and Bi) system at cathodic potentials preventing formation/coordination with oxygenated species from the electrolyte that are discussed to lower activity and selectivity for CO₂RR to formate.

Overall, the research is well designed and presented in the manuscript. The experiments seem to have been carried out with care. Electrocatalytic characterisation, material synthesis and ex situ characterisation seem properly conducted and clearly presented in the manuscript.

The major flaws lie on the mechanism part need to be addressed before reconsideration for a publication in Nature Communications could be considered.

Major comments:

1.1. The authors mention on line 108 that the defect which Bi catalysts can easily undergo oxidation by oxygenated species point the author should clarify if this is their results & interpretation (from the calculations) or if this has been reported in the literature. I am not aware any works.

1.2. The currents densities are reported, e.g., Figure 3b. How was the active surface area determined? Which areas are the authors referring to?

1.3. How the VC is bound to the on the BiS-surface?

Important aspects to understand the role of the VC layer should be more clearly presented:

1.4. The authors noted that after electrolysis, the BS/VC system exhibited a porous structure. Is the surface area of the BS/VC increased during the electrocatalysis?

1.5. The fate of the VC layer during and post-electrolysis is an important aspect. On line 219, the authors stated that the VC coating layer is less uniform after electrolysis. The authors need to add the details of the electrolysis (it's hidden in the SI), so the reader can compare, e.g., electrolysis conditions & time.

1.6. The FT-EXFAS data is not fully clear to this reviewer. This reviewer agrees with the authors, that the VC seems to affect the Bi(0) formation and the morphology of the active catalyst. However, on line 258, the authors stated that the Bi foil (Fig. S26) shows mainly Bi-Bi signals in the FT-EXAFS correlation spectrum. Why does the Bi foil lack the Bi-O signals at -1.4V, while the BiS shows Bi-O signals? It is contrary to the statement on line 108. Moreover, the BiS system also shows different Bi-Bi correlation

signals, why is it so and what does this mean? Should this also not imply coordinative non-usual saturated sites?

1.7. The application of ATR-SEIRAS is much appreciated, but the presentation and description of the data must be more carefully done. It is noted that the VC covered BiS seems to show deviating potential-dependent ATR-SEIRAS spectra. However, the assignment of bands seems not to be correct to this reviewer. The authors stated on line 279 (to 284) that the signals at 1405 and 1640 cm^{-1} are coming from the carboxylate. 1640 cm^{-1} is clearly the H-O bending mode and is known to arise when going to cathodic potentials. COO^- should be expected at different and distinctly lower frequencies, e.g., 1560 cm^{-1} and 1380 cm^{-1} for the symmetric and asymmetric stretching mode, respectively. Also, with rising of the 1640 cm^{-1} , the authors also noted the accompanied rising of bands above 3200 cm^{-1} , which typically is the H-O stretching mode, as they also noted on line 285.

1.8. Which potential was used as reference potential for calculation of the ATR-SEIRAS difference spectra presented in Figure 5?

Minor

2.1 The separation of the results and discussion part seems not useful. The discussion part is only half a page.

2.2 This reviewer suggest calculation of Tafel plots for comparing the Tafel slopes of the BiS-VC and BiS (and other Bi) to catch any changes in the mechanistic behaviour.

2.3 The authors mention Raman measurements in the experimental part, but no results obtained with Raman are presented.

2.4 The authors have used Au and deposited the catalyst ink on top. This has been done before and seem to yield good results. The authors are advised to cite given works, e.g., 10.1038/s41467-020-15141-y.

Reviewer #3 (Remarks to the Author):

Comments for the authors

In this study, Dai et.al report a Bi₃S₂ nanowire-ascorbic acid hybrid catalyst for the high rate production of formic acid (partial current density of 910 mA/cm²) from electrochemical CO₂ conversion. The authors attribute the high reaction rates achieved to the trapping of poisoning hydroxyl groups by VC layer and defective sites generated in the hybrid catalyst. Detailed material characterization and operando techniques are performed which is commendable. Although the design strategy and the performance are quite interesting, the work lacks detailed analysis and clarity in a few areas. Hence, it may not be suitable for publication in Nature Communications journal.

Comments:

1. The authors report a partial current density of 910 mA/cm² for formate production. While potentials are reported vs RHE, it is advised to also report the total cell voltage and energy efficiency for any claim of industrial relevance.
2. In Line 132: The authors claim an FE of 80 % for formate. What were the remaining products? This should be reported.
3. In Line 187: The authors report a current density of 1.26 A/cm². How long was this experiment performed? For industrial relevance, a current density of 200 mA/cm² and at least 100 hours of operation are required. Stability tests performed in Figure 6 are performed at 50 or 100 mA/cm².
4. The authors report the use of a Sustainion AEM for flow cell experiments at higher current densities. In this system, water crossover (due to OH⁻, CO₃²⁻ and HCO₃⁻ ions) leads to a variation in electrolyte volume resulting in an overestimation of liquid products. Did the authors account for this?
5. The authors report the use of a Pt counter electrode for H-cell experiments. However, Pt is not a recommended choice due to the issue of Pt dissolution especially over long running hours. The authors should comment on this and verify that this may not affect their results.
6. Line 366: The authors report the use of Ag/AgCl as reference electrode in 1 M KOH electrolyte. This is not advised because silver oxide/hydroxide formation in KOH is a well-known phenomenon in electrochemistry. Ag/AgCl is only recommended for neutral electrolytes. A Hg/HgO reference electrode is hence recommended to avoid this issue. (See: Niu, Siqi, et al. "How to reliably report the overpotential of an electrocatalyst." ACS Energy Letters 5.4 (2020): 1083-1087).
7. While detailed material characterization for Bi/VC catalyst (SEM, HRTEM) on carbon paper are reported for studies in H-cells, details on how this coating was performed on gas diffusion layers for flow cells are missing. This is especially relevant at higher current densities, since the GDE has a porous structure and reporting details on how such a stable coating was achieved are beneficial.

8. Recent studies on formate production have shown that carbonate species as reaction intermediate for CO₂RR to formate in Bi catalysts (<https://pubs.acs.org/doi/10.1021/acsomega.7b00437>). The authors should report if BS/VC catalysts create any such intermediates.
9. At higher current densities, formate production is often hampered by sluggish water dissociation kinetics. Previous studies have attributed catalyst design strategies to increase formate production by increasing water dissociation kinetics. However, no discussion on this aspect is provided. It would be beneficial if discussion on how BS/VC affect water dissociation kinetics is discussed for more clarity.
10. Some reports on In based catalysts producing formate from CO₂RR have reported that surface hydroxyl activates CO₂ reduction. (<https://pubs.acs.org/doi/10.1021/la501245p>). However, in this work the authors claim trapping of hydroxyl species increases formate production. This is contradictory and the authors need to clarify and provide explanation for the observed differences.
11. Electrochemical active surface area for BS/VC and BS must be calculated to show ECSA normalized current densities.
12. NMR spectra and a sample calculation for quantification of formate in flow cells at high current densities must be reported.
13. At higher current densities, how much of CO₂ fed reacted with OH⁻ ions generated during the reaction? Single pass CO₂ utilization efficiency must also be reported since it's one of the important performance metrics.
14. Flooding of the GDE due to capillary pressure differences especially with KOH electrolyte is an issue. How did the authors circumvent this issue at 1.26 A/cm² reported current density?
15. If poisoning by hydroxyl groups is the main hindrance for reaching higher reaction rates, can this strategy be applied for other formate producing catalysts such as Sn or In? This should be clarified and if so, preferably an experiment using Sn/VC catalyst should be performed at least in H-cells.
16. Line 76: "CORR process". Typo in CO₂ instead of CO?

Point-to-Point Responses to Reviewers' Comments

(Manuscript ID: NCOMMS-23-05199-T-R1): Surface passivation for highly active, selective, stable, and scalable CO₂ electroreduction

Reviewer: 1

Comments to the Author

The manuscript by Zhu et al. examines how chemical functionalization of the catalyst surface can be used to tailor the microenvironment near the active site. This study shows that vitamin c functionalization of a bismuth catalyst (wherein defect sites provide catalytically active sites) enhances the selectivity toward formic acid and extends the durability of the catalyst.

The authors provide a comprehensive characterization of the structure and surface chemistry of the defect-rich bismuth catalyst including, TEM, XPS, and FTIR.

The study is based on a close integration of experiment and computational simulation (i.e., binding energies from DFT).

Overall, the results are interesting and the mechanistic interpretation is, in large part, supported by the provided experimental and computational evidence. The authors convincingly show that oxidation of the defective Bi sites reduced the formate selectivity and that this effect can be mitigated by the functionalization with vitamin C (VC)

However, there are also several aspects that should be revised to improve the quality and impact of the study (I.e., accept after major revisions to address the points outlined below)

Response to Reviewer 1:

We thank the reviewer for the approbatory comments on our manuscript. We welcome the opportunity to address and clarify the issues raised in the reviewer's comments, which we believe to substantially strengthen our manuscript. Below, please find our responses to the points raised in the Reviewer.

Reviewer Comments:

Comment 1. The authors point to several recent studies which showed that chemical modification of the microenvironment can impact adsorption behavior. While previous studies as well as the current study show an enhancement effect it seems that there should be an underlying trade-off. On the one hand functionalizing (i.e., passivating) the catalyst surface affords more control over chemical functionality (e.g., hydrophilicity to control adsorption of intermediates and spectator ions); on the other hand, if the functionalizing species is bound to the surface (active site), then this site is unavailable to activate the heterogeneous catalytic reaction. It would be helpful to discuss this trade-off and associated design principles to guide the optimal balance of functionality and activity.

Respond to comment-1: We thank the reviewer for the insightful comment. As the reviewer said, there is a trade-off on surface molecular modification catalysts. If the molecular/inomer

bound to the active sites, the catalytic reaction can not proceed on these sites. Thus, it would be better if the molecular just as a coating rather than bound with active sites. Up to now, most molecular/inomers reported in high impact journal are used to adjust the local reaction environment (Nat. Mater. 18, 1222–1227 (2019); Nat Catal 3, 75–82 (2020); Nat Energy 5, 478–486 (2020); Science 367, 661–666 (2020)). In our work, the VC acts as a coating layer to regulate the microenvironment of Bi-based catalysts and does not bond to the surface. We have added the discussion in the main text. We believe these revisions according to the reviewer's comment can substantially strengthen the revised manuscript.

Main text:

Page 4: Moreover, it is crucial to manage the adsorption configuration of the molecular layer on catalysts, as it can obstruct the adsorption of reactants to active sites if the molecules bind to those sites. The molecular layer adheres to the catalyst surface through electrostatic forces, generating a confined reaction space that enhances the activation behavior of the reaction intermediates at the interface between the catalyst and the molecular layer, thus offering more reliable feasibility.³³⁻³⁵

Comment 2. The authors present a series of DFT calculations of binding energies on bismuth surfaces with varying defect densities. These calculations provide important initial insights into the role of defects which are subsequently examined in experimental studies. One of the most important aspect of these calculations is the strong binding of the ascorbic acid anti-oxidant to the catalyst surface. It would be helpful to provide more information on the binding configuration on the defect-rich bismuth surface. How and where is the VC bound? Figure 1d compares binding energies, but doesn't specify the defect type or where/how the VC is bound. Is VC chemically bound or adsorbed?

Respond to comment-2: We thank the reviewer for the valuable comment. Actually, we showed the strong binding of the ascorbic acid to O₂ and OH⁻ species in Figure 1d, rather than the binding to catalyst surface. The VC is physically adsorbed on BS surface by electrostatic forces because there is no Bi–O bond observed in O 1s XPS spectra in Figure S21. Besides, the thickness of VC layer show no great change with the adjustment of VC solution concentration since the effective range of electrostatic force is within a short range (see the respond to comment 3). We have added the discussion in the main text. We believe these revisions suggested by the reviewer substantially strengthen the revised manuscript.

Main text:

Page 11: It is worth noting that there is no Bi–O bond observed. Considering that the thickness of VC is not affected by the concentration of VC solution, we believe that VC is mainly present on the BS surface in the form of physical adsorption through the electrostatic interaction.

Comment 3. The effect of adding VC to maintain high formic acid selectivity is clearly evident from the presented results. However the catalyst design principles and impact of the paper could be significantly enhanced by further quantifying this effect. Beyond the A/B comparison of (A with VC , B without VC) it would be helpful to quantify this effect.

- a. What happens if the VC coverage is increased / decreased?
- b. What happens with other surface functionalizations? (i.e., species that also bind to the defect-rich Bi surface but don't have anti-oxidant effects?)

Addressing these questions would be very helpful and would also help to address the trade-off between functionality and activity described above.

Respond to comment-3: We thank the reviewer for the constructive comment.

- a. We tried to adjust the VC solution concentration when we synthesized the BS/VC. As shown in Figure S20, when we reduce the VC concentration to 0.25 M, the thickness of VC coating reduce to 4 nm. However, when the VC concentration increase to 1 M and 2 M, the thickness of VC coating show negligible change, meaning that the coating thickness of VC has reached its limit under 0.5 M VC. Thus, we believe that the thickness of VC layer is mainly controlled by the range of electrostatic forces of BS, and it may not have obvious effect on the activity and selectivity of CO₂RR. We have added the discussion in the main text.

Figure S20. HAADF-STEM image of BS/VC with different VC concentration. (a) 0.25 M, (b) 1 M, (c) 2 M.

Main text:

Page 10: Through adjusting the VC solution concentration, we found that the VC layer show no great change in thickness (**Supplementary Fig. 20**).

- b. We introduced four types of molecular species on the surface of BS and investigate the effect on activity and formate selectivity. The results show that with other surface functionalizations that don't have antioxidant effect, the CO₂RR activity and formate

selectivity will not be enhanced, but weakened. It further implies the importance of antioxidant of a surface coating. We have added this discussion in the main text.

Figure S34. (a) The structure formula of 4-Nitrothiophenol. (b) FT-IR analysis of BS/I, and I. I represent the 4-Nitrothiophenol. (c) The LSV curve of BS/I in flow-cell system with 1 M KOH electrolyte. (d) The selectivity of HCOO⁻, H₂, and CO under different applied potential.

Figure S35. (a) The structure formula of 5,6-Dimethylbenzimidazole. (b) FT-IR analysis of BS/II, and II. II represent the 5,6-Dimethylbenzimidazole. (c) The LSV curve of BS/II in flow-cell system with 1 M KOH electrolyte. (d) The selectivity of HCOO⁻, H₂, and CO under different applied potential.

Figure S36. (a) The structure formula of 9-Anthracenecarboxylic acid. (b) FT-IR analysis of BS/III, and III. III represent the 9-Anthracenecarboxylic acid. (c) The LSV curve of BS/III in flow-cell system with 1 M KOH electrolyte. (d) The selectivity of HCOO^- , H_2 , and CO under different applied potential.

Figure S37. (a) The structure formula of 1-Dodecanethiol. (b) FT-IR analysis of BS/IV, and IV. IV represent the 1-Dodecanethiol. (c) The LSV curve of BS/IV in flow-cell system with 1 M KOH electrolyte. (d) The selectivity of HCOO⁻, H₂, and CO under different applied potential.

Main text:

Page 20: To further examine the distinctiveness of VC, we employed four types of molecular species (**I**: 4-Nitrothiophenol; **II**: 5,6-Dimethylbenzimidazole; **III**: 9-Anthracenecarboxylic acid; **IV**: 1-Dodecanethiol) on the surface of BS (**Supplementary Fig. 34-37**). None of these four molecules have antioxidant capacity, but **I** and **IV** possess hydrophobic groups. Compared to VC coating, the introduction of molecules **I** - **IV** resulted in a reduction in both current density and the selectivity of formate, which was even lower than that of BS. The decline in current density can be attributed to the poor conductivity of molecules. The formate selectivity of BS/**I** and BS/**IV** was noticeably higher than that of BS/**II** and BS/**III**, but slightly lower than that of BS. This is because the hydrophobic nature of **I** and **IV** which facilitated the adsorption

of CO₂ onto the catalysts. These results further highlight the importance of antioxidant to Bi-based catalysts.

Comment 4. Figure 3 shows remarkably high current densities as high as -1.2 A/cm², yet the extended operating time measurements in Figure 6 show only 0.1 or 0.05 A/cm². How does the durability of the electrolyzer depend on the current density? Even if the durability at higher current density is reduced, this would still be an important and interesting result to share.

Respond to comment-4: We thank the reviewer for the valuable comment. We obtained a high current density of -1.2 A cm⁻² with 1.0 cm² electrode area in alkaline flow cell test. When we conducted the durability test in a solid-state reactor, we used the electrode with 4.0 cm² electrode area, so the current density of 0.1 and 0.05 A cm⁻² were actually obtain at 0.4 and 0.2 A current. We also conducted the durability test of BS/VC under 0.2 A cm⁻² (0.8 A), and the formate selectivity show quickly reduced trend. It may be ascribed to the unstable adsorption of VC layer under the high current, and thus expose the active sites to the hydroxyl generated during the CO₂RR, leading to the decrease in activity and selectivity. We have added the discussion in the main text. We believe these revisions as suggested by reviewer have substantially strengthen the revised manuscript.

Figure S41. Long-term production of pure formic acid from the BS/VC at current densities of 200 mA cm⁻², in the all-solid-state reactor.

Main text:

Page 22: However, when the current density increases to 200 mA cm⁻² (800 mA cell current), the formate selectivity shows rapid decline (**Supplementary Fig. 41**), which may be ascribed to the unstable adsorption of VC layer.

Comment 5. Figure 4 includes additional structure analysis including tomographic reconstruction of the nanowire. The data are nice, but not that critical to the main theme of the rest of the paper.

Respond to comment-5: We thank the reviewer for the thoughtful comment. In Figure 1, we have discussed and showed the importance of defect Bi site to the selectivity of CO₂RR. Without the coating of VC layer, the BS nanowires would convert into Bi nanosheets under CO₂RR conditions. In contrast, the VC layer can inhibit the complete reconstruction of BS nanowires and obtain highly porous Bi nanowires. Obviously, the porous Bi nanowires possess more defect sites than Bi nanosheets. Thus, the reconstruction restriction effect of VC ensures the enhanced activity of CO₂RR and the antioxidant effect improves the stability of high activity.

Comment 6. Figure 5g provides a nice schematic illustration of the mechanistic interpretation of the role of defect sites and how VC functionalization impacts the microenvironment near the active site. The bottom half of Figure 5g shows two different reaction environments; presumably the outside and inside of the hollow wire. I don't think the interpretation about the reaction on the hollow core of the wire is (a) necessary or (b) supported by the presented data, so I suggest to just focus on the outside surface.

Respond to comment-6: We thank the reviewer for the thoughtful comment. We have revised the schematic according to the reviewer's suggestion. The new schematic is shown in Figure 5g.

Figure 5g. Schematic illustration of the hydroxyl trapping by VC passivate layer and ensure a smooth CO₂ reduction at the defective sites.

Comment 7. On pg 4, the authors state that "... catalytic reactions occur mainly on the catalyst surface, surface modification can significantly regulate the process of catalytic reactions." I would say that by definition heterogeneous catalytic reactions occur at the surface (active site),

so the statement about 'mainly' should be removed, unless I misinterpreted the intended message.

Respond to comment-7: We thank the reviewer for the helpful comment. We have revised the corresponding description accordingly in the main text.

Main text:

Page 4: Since catalytic reactions occur on the catalyst surface, surface modification can significantly regulate the process of catalytic reactions.

Reviewer: 2

Comments to the Author

The authors report a Bi₃S₂ catalyst interface-modified with ascorbic acid for CO₂RR to formate. The manuscript introduces the hybrid catalyst delivering outstanding catalytic activity. The hybrid catalyst showed very high FE at impressive ampere current densities with stability over 120hrs (in a solid-electrolyte reactor system). The presented system competes well with state-of-the-art Bi-based electrocatalysts reported.

The origin of the catalytic properties is then discussed on a mechanistic basis. The authors have used SEIRA spectroelectrochemistry to monitor the BiS-VC surface under potential. It is concluded that the surface functionalisation achieved by layering ascorbic acid is claimed to stabilise the BiS (and Bi) system at cathodic potentials preventing formation/coordination with oxygenated species from the electrolyte that are discussed to lower activity and selectivity for CO₂RR to formate.

Overall, the research is well designed and presented in the manuscript. The experiments seem to have been carried out with care. Electrocatalytic characterisation, material synthesis and ex situ characterisation seem properly conducted and clearly presented in the manuscript.

The major flaws lie on the mechanism part need to be addressed before reconsideration for a publication in Nature Communications could be considered.

Response to Reviewer 2:

We thank the reviewer for the approbatory comments on our manuscript. We welcome the opportunity to address and clarify the issues raised in the reviewer's report and believe that the revisions substantially strengthen our manuscript. Our responses to the points raised in the report are below.

Reviewer Comments:

Comment 1. The authors mention on line 108 that the defect which Bi catalysts can easily undergo oxidation by oxygenated species point the author should clarify if this is their results

& interpretation (from the calculations) or if this has been reported in the literature. I am not aware any works.

Respond to comment-1: We thank the reviewer for the valuable comment. We calculated the binding energy of O₂ and OH⁻ on intact Bi (p-Bi) and single Bi-vacancy (sv-Bi) defect. We found that the sv-Bi shows a low binding energy to O₂ and OH⁻, meaning that it is easy to be oxidized by them. We realize that our expression is not clear enough that may cause some confusions. We have added the discussion in the main text. We believe these revisions to avoid the unnecessary confusions.

Main text:

Page 6: The high activity of defect sites likely leads to binding with species in the air or in the electrolyte solution, resulting in inactivation. From our calculations, we found that the sv-Bi shows a low binding energy to O₂ and OH⁻ (Fig. 1d), leading to a decrease in the Bi vacancy density and weak *OCHO adsorption, and hence the catalytic performance degradation (Fig. 1e).

Comment 2. The currents densities are reported, e.g., Figure 3b. How was the active surface area determined? Which areas are the authors referring to?

Respond to comment-2: We thank the reviewer for the valuable comment. We use catalysts loaded on a gas diffusion electrode as the work electrode, and assembled it into an alkaline flow cell electrolyzer, as shown in Figure 3a. The current density is determined by the collected current divided by the electrode area which expose to the electrolyte.

Comment 3. How the VC is bound to the on the BiS-surface?

Important aspects to understand the role of the VC layer should be more clearly presented:

Respond to comment-3: We thank the reviewer for the insightful comment. We believe that the VC is physically adsorbed on BS surface by electrostatic forces because there is no Bi–O bond observed in the O 1s XPS spectra in Figure S21. Besides, the thickness of the VC layer shows not so much change with the VC solution concentration since the effective range of electrostatic force is limiting factor for the VC adsorption through the electrostatic interaction. We have added this discussion in the revised main text. We believe these revisions substantially strengthen the revised manuscript.

Figure S20. HAADF-STEM image of BS/VC with different VC concentration. (a) 0.25 M, (b) 1 M, (c) 2 M.

Comment 4. The authors noted that after electrolysis, the BS/VC system exhibited a porous structure. Is the surface area of the BS/VC increased during the electrocatalysis?

Respond to comment-4: We thank the reviewer for the constructive comment. To investigate the hypothesis, we conducted the electrochemical surface area test. The results show that the active area of BS and BS/VC both increase in a similar manner during the CO₂RR. Thus, the increase surface area is not solely responsible for the enhanced activity and selectivity on BS/VC. We have added the discussion in the revised main text. We believe these revisions according the the reviewer's comments substantially strengthen our revised manuscript.

Figure S27. The CV curves with different scan rate under the potential of 0.75 V – 0.85 V of BS, BS/VC, BS-after, and BS/VC-after. To avoid the affect of porous electrode (gas diffusion electrode) and electrolyte flow, we conducted the electrochemical surface area (ECSA) test by dropping the catalysts ink on Indium Tin Oxide (ITO) coated glass and run the CV in 1 M KOH without flow.

Figure S28. Δj at 0.8 V vs. RHE as a function of the scan rate to evaluate C_{dl} .

Main text:

Page 16: An electrochemical active surface area (ECSA) test was conducted on catalysts before and after electrolysis to investigate the active area change during CO₂RR (**Supplementary Fig. 27-28**). The results show that with VC coating, the BS/VC exhibited a slight decline in active area, which could be attributed to the nanowire aggregation induced by the surface-attached molecules. After electrolysis, the reconstruction of nanowires led to an increase of the active area, whereas the active area of the BS/VC-after remained lower than that of the BS-after. The similar trend for the BS and BS/VC electrodes indicates that the increase in active area is not solely responsible for the enhanced activity and selectivity observed for the BS/VC.

Comment 5. The fate of the VC layer during and post-electrolysis is an important aspect. On line 219, the authors stated that the VC coating layer is less uniform after electrolysis. The authors need to add the details of the electrolysis (it's hidden in the SI), so the reader can compare, e.g., electrolysis conditions & time.

Respond to comment-5: We thank the reviewer for the helpful comment. Sorry for missing some necessary experiment details. We have added the discussion in the revised main text and supporting information. We believe these revisions substantially strengthen the revised manuscript.

Main text:

Page 14: The samples for ex-situ TEM were collected by dropping the catalysts on Indium Tin Oxide (ITO) coated glass and proceed the CO₂RR at 200 mA cm⁻² for 24 hours in CO₂-saturated 0.5 M KHCO₃.

Figure S24. SEM images of the BS-derived Bi (a) BS/VC-derived Bi/VC on carbon paper after 24 hours electrolysis in 0.5 M KHCO₃.

Comment 6. The FT-EXFAS data is not fully clear to this reviewer. This reviewer agrees with the authors, that the VC seems to affect the Bi(0) formation and the morphology of the active catalyst. However, on line 258, the authors stated that the Bi foil (Fig. S26) shows mainly Bi-Bi signals in the FT-EXAFS correlation spectrum. Why does the Bi foil lack the Bi-O signals at -1.4V, while the BiS shows Bi-O signals? It is contrary to the statement on line 108. Moreover, the BiS system also shows different Bi-Bi correlation signals, why is it so and what does this mean? Should this also not imply coordinative non-usual saturated sites?

Respond to comment-6: We thank the reviewer for the insightful comment. In Figure S31 (the original Figure S26), the XAS of Bi foil was not collected under in-situ electrochemical condition, and thus no Bi-O was observed. The difference Bi-Bi coordination signals between BS and BS/VC is due to the structure reconstruction under a high negative applied potential. Without the VC layer, the Bi atoms quickly reorganized into a complete lattice, so no perturbed coordination structure was detected. The reconstruction range of BS/VC-derived Bi is limited by the VC coating, leading to the higher disturbance in structure, thus a less unusual Bi-Bi coordination. The corresponding description is shown in revised main text.

Main text:

Page 17: At -1.4 V, the Bi units in the BS and BS/VC were totally converted into Bi, and the difference is that the BS-derived Bi exhibited a similar coordination structure to the Bi foil, while the BS/VC-derived Bi showed a significant coordination structure perturbation. Without the VC layer, the Bi atoms quickly reorganized into a complete lattice, so no perturbed coordination structure was detected. The BS/VC-derived Bi exhibited a less unusual Bi–Bi coordination due to large perturbations at more negative potentials, including the dynamic reconfiguration processes and diffusion of products. Results from the corresponding wavelet transform (**Supplementary Fig. 31**) show that the BS/VC contains only the Bi–Bi coordination at -1.4 V, suggesting that the perturbed signal came from the dynamically reconfigured Bi atoms and not from other coordination structures.

Comment 7. The application of ATR-SEIRAS is much appreciated, but the presentation and description of the data must be more carefully done. It is noted that the VC covered BiS seems to show deviating potential-dependent ATR-SEIRAS spectra. However, the assignment of bands seems not to be correct to this reviewer. The authors stated on line 279 (to 284) that the signals at 1405 and 1640 cm^{-1} are coming from the carboxylate. 1640 cm^{-1} is clearly the H-O bending mode and is known to arise when going to cathodic potentials. COO^- should be expected at different and distinctly lower frequencies, e.g., 1560 cm^{-1} and 1380 cm^{-1} for the symmetric and asymmetric stretching mode, respectively. Also, with rising of the 1640 cm^{-1} , the authors also noted the accompanied rising of bands above 3200 cm^{-1} , which typically is the H-O stretching mode, as they also noted on line 285.

Respond to comment-7: We thank the reviewer for the thoughtful comment. The frequency range for the symmetric and asymmetric stretching mode of COO^- should be located at the range of 1430 - 1350 cm^{-1} and 1620 - 1540 cm^{-1} , respectively. And as the reviewer said, the frequency of 1640 cm^{-1} should be the H-O bending of H_2O . Considering the two peaks of the COO^- present at the same time, we believe the wide peak around 1640 cm^{-1} should be divided into two peaks, the H-O bending of H_2O and asymmetric stretching of COO^- . We have revised the figure and added this discussion into the revised main text. We believe these revisions made by following the reviewer's comments substantially strengthen our revised manuscript.

Figure 5e. In-situ ATR-SEIRAS spectra of BS

Figure 5f. In-situ ATR-SEIRAS spectra of BS/VC.

Main text:

Page 19: As shown in **Fig. 5e-f**, the two upward enhancement bands at 1395 and around 1600 – 1620 cm^{-1} are attributable to the symmetric and asymmetric stretching vibration of COO radicals with a difference between the two bands greater than 200 cm^{-1} , indicating that the COO radicals are bound to the metal sites through a monodentate ligand, i.e., M-OCO.

Since CO_2RR is a proton-consuming reaction, a large amount of OH^- was generated during the reaction, showing absorption peaks of 1630 – 1650 and 3600 – 3200 cm^{-1} in the IR spectra.

Comment 8. Which potential was used as reference potential for calculation of the ATR-SEIRAS difference spectra presented in Figure 5?

Respond to comment-8: We thank the reviewer for the helpful comment. The potential titles in the spectra of in-situ ATR-SEIRAS are referenced to reversible hydrogen electrode. We have added the corresponding note in the figure caption in our revised manuscript.

Comment 9. The separation of the results and discussion part seems not useful. The discussion part is only half a page.

Respond to comment-9: We thank the reviewer for the helpful comment. Sorry for the misleading. The discussion part in our manuscript actually corresponds to the conclusions part. We have changed the section title of **Discussion** into **Conclusions**.

Comment 10. This reviewer suggest calculation of Tafel plots for comparing the Tafel slopes of the BiS-VC and BiS (and other Bi) to catch any changes in the mechanistic behaviour.

Respond to comment-10: We thank the reviewer for the valuable comment. As suggested by the reviewer, we calculated the Tafel slopes for BS and BS/VC and indeed found that they showed different reaction mechanisms. the Tafel slope of BS is 261 mV dec^{-1} , close to 200 mV dec^{-1} , indicating a high barrier for the adsorption of CO_2 to active sites. In contrast, the Tafel slope of BS/VC is 125 mV dec^{-1} , indicating the high barrier for charge transfer to form $^*\text{CO}_2^-$. This finding further highlights the impact of VC on trapping or isolating hydroxyl groups. We have added this discussion in the revised main text and supporting information. We believe these revisions according to the reviewer's suggestion substantially strengthen the revised manuscript.

Figure S33. Tafel slope of BS and BS/VC.

Main text:

Page 20: The Tafel slope also provides valuable information for investigation of the reaction mechanism. As shown in **Supplementary Fig. 33**, BS has a Tafel slope of 261 mV dec^{-1} , close to 200 mV dec^{-1} , indicating that a high barrier for the adsorption of CO_2 to active sites.⁵⁰⁻⁵² It is consistent with the poison of hydroxyl to active sites and prevent the adsorption of CO_2 molecules. A Tafel slope of 125 mV dec^{-1} for BS/VC is indicative of a high barrier for charge transfer to form $^*\text{CO}_2^-$ with no limitation for CO_2 adsorption.⁵²⁻⁵⁴ This further demonstrates the effect of VC on the trapping or isolation of hydroxyl groups.

Comment 11. The authors mention Raman measurements in the experimental part, but no results obtained with Raman are presented..

Respond to comment-11: We thank the reviewer for the helpful comment. Sorry for our carelessness. We have remove the Raman measurements in experimental part.

Comment 12. The authors have used Au and deposited the catalyst ink on top. This has been done before and seem to yield good results. The authors are advised to cite given works, e.g., 10.1038/s41467-020-15141-y.

Respond to comment-12: We thank the reviewer for the valuable comment. This is of great help for our IR data analysis. We have cited the given paper as the Ref. 47 in our revised manuscript.

47 Zhong, H. *et al.* Synergistic electroreduction of carbon dioxide to carbon monoxide on bimetallic layered conjugated metal-organic frameworks. *Nat. Commun.* **11**, 1409 (2020).

Reviewer: 3

Comments to the Author

In this study, Dai et.al report a Bi_3S_2 nanowire-ascorbic acid hybrid catalyst for the high rate production of formic acid (partial current density of 910 mA/cm^2) from electrochemical CO_2 conversion. The authors attribute the high reaction rates achieved to the trapping of poisoning

hydroxyl groups by VC layer and defective sites generated in the hybrid catalyst. Detailed material characterization and operando techniques are performed which is commendable. Although the design strategy and the performance are quite interesting, the work lacks detailed analysis and clarity in a few areas. Hence, it may not be suitable for publication in Nature Communications journal.

Response to Reviewer 3:

We thank the reviewer for the pertinent comments, which are very helpful to further improve our manuscript. We welcome the opportunity to address and clarify the issues raised in the reviewer's report and believe that the revisions according to the reviewer's comments substantially strengthen our manuscript. Our responses to the points raised by the reviewer are below.

Reviewer Comments:

Comment 1. The authors report a partial current density of 910 mA/cm² for formate production. While potentials are reported vs RHE, it is advised to also report the total cell voltage and energy efficiency for any claim of industrial relevance.

Respond to comment-1: We thank the reviewer for the valuable comment. The partial current density of 910 mA cm⁻² for formate production was reported in alkaline flow-cell, which was tested in three-electrode system, so there was no cell voltage can be reported. In the solid-state reactor test, we use two-electrode system to collect the data, and we added the total cell voltage and energy efficiency of formic acid as a function of current density. We have added this discussion in our revised main text. We believe these revisions suggested by the reviewer substantially strengthen the revised manuscript.

Figure S40. The total cell voltage and energy efficiency of formic acid as a function of current density.

Main text:

Page 21: The energy efficiency of formic acid as a function of current density reveals that the BS/VC achieved an energy efficiency of 33%, and 22% can be maintained at high current density (**Supplementary Fig. 40**).

Page 26: In solid-state reactor, the formic acid energy efficiency is calculated as follows:

$$EE_{\text{full-cell}} = \frac{(1.23 + (-E_{\text{formic acid}})) \times FE_{\text{formic acid}}}{-E_{\text{full-cell}}}$$

where $E_{\text{full-cell}}$ is the cell voltage applied in the solid-state reactor, and the $E_{\text{formic acid}} = -0.2$ V.⁶²

Comment 2. In Line 132: The authors claim an FE of 80 % for formate. What were the remaining products? This should be reported. (H₂ CO)

Respond to comment-2: We thank the reviewer for the helpful comment. The remaining products are H₂ and CO, and the corresponding faradaic efficiency are given in Figure 2b. We have added more discussions in the revised main text. We believe these revisions substantially strengthen the revised manuscript.

Main text:

Page 8: Around 80% FE could be achieved at a potential range of 0.6 V and the formate was the only liquid product (**Supplementary Fig. 9**), while the remaining products are mainly H₂ with little CO.

Comment 3. In Line 187: The authors report a current density of 1.26 A/cm². How long was this experiment performed? For industrial relevance, a current density of 200 mA/cm² and at least 100 hours of operation are required. Stability tests performed in Figure 6 are performed at 50 or 100 mA/cm².

Respond to comment-3: We thank the reviewer for the valuable comment. The current density of 910 mA cm⁻² was collected in alkaline flow-cell system within 1 hour. As we all know, the gas diffusion electrode suffer from flooding in alkaline flow-cell system, so that it is hard to examine the stability of catalysts. Therefore, we conducted the durability test in a solid-state reactor. We used the electrode with 4.0 cm² electrode area, so the current density of 0.1 and 0.05 A cm⁻² were actually obtain at 0.4 and 0.2 A current. As shown in Fig. 6, a stable operation over 120 hours with the HCOOH selectivity maintained above 80% (**Fig. 6c**) was achieved at 50 mA cm⁻² (200 mA cell current). Even at a current density of 100 mA cm⁻² (400 mA cell current), the cell voltage and product selectivity remained stable for more than 80 hours. As the reviewer suggested, we also conducted the durability test of BS/VC under 0.2 A cm⁻² (0.8 A cell current), and found that the formate selectivity show a quickly reduced trend. It may be ascribed to the unstable adsorption of VC layer under the high current, and thus expose the active sites to the hydroxyl generated during the CO₂RR, leading to the decrease in activity and selectivity. **This paper is focused on communicating the fundamentally new strategy and we feel because of its novelty and importance may spur unforeseen interests by other groups worldwide to further study and to improve the stability for practical applications. So, it may be unnecessary, though important, to over emphasize the stability under harsh conditions from the present work.** We have added the relevant discussion in our revised main text and hope that these revisions will make the situation clear.

Figure S41. Long-term production of pure formic acid from the BS/VC at current densities of 200 mA cm^{-2} , in the all-solid-state reactor.

Main text:

Page 22: However, when the current density increases to 200 mA cm^{-2} (800 mA cell current), the formate selectivity shows rapid decline (**Supplementary Fig. 41**) due, most probably, to an unstable adsorption of the VC layer.

Comment 4. The authors report the use of a Sustainion AEM for flow cell experiments at higher current densities. In this system, water crossover (due to OH^- , CO_3^{2-} and HCO_3^- ions) leads to a variation in electrolyte volume resulting in an overestimation of liquid products. Did the authors account for this?

Respond to comment-4: We thank the reviewer for the insightful comment. When we used the Sustainion AEM membrane in alkaline flow-cell system, the anion crossover (OH^- , CO_3^{2-} , HCO_3^- , and HCOO^-) might happen. Considering the flooding of electrode, we didn't run the test over 1 hour in alkaline flow-cell system, so we didn't observe the change of electrolyte volume for the anode and cathode.

Comment 5. The authors report the use of a Pt counter electrode for H-cell experiments. However, Pt is not a recommended choice due to the issue of Pt dissolution especially over long running hours. The authors should comment on this and verify that this may not affect their results.

Respond to comment-5: We thank the reviewer for the kind and thoughtful comment. Indeed, due to the high applied potential of oxygen evolution on Pt electrode, the Pt may dissolve in a long time test. Firstly, we did the CO_2RR test in three-electrode system, so the activity change of oxygen evolution caused by a small amount of Pt dissolution have negligible effect on the

activity and selectivity of CO₂RR. Besides, we did the test at a low current density within several hours, and thus the dissolution of Pt is little. If the dissolved Pt crosses over the membrane and deposits on the cathode surface, the activity and selectivity of CO₂RR may be reduced. However, within our several hours test, the membrane can block most of the dissolved Pt, if any. Moreover, when we tested the BS catalysts in alkaline flow-cell system with Ni foam as anode, the formate selectivity show similar trend compared to that in the H-cell. We believe that the Pt dissolution have negligible effect on the CO₂RR activity of BS electrode in the H-cell.

Comment 6. Line 366: The authors report the use of Ag/AgCl as reference electrode in 1 M KOH electrolyte. This is not advised because silver oxide/hydroxide formation in KOH is a well-known phenomenon in electrochemistry. Ag/AgCl is only recommended for neutral electrolytes. A Hg/HgO reference electrode is hence recommended to avoid this issue. (See: Niu, Siqi, et al. "How to reliably report the overpotential of an electrocatalyst." ACS Energy Letters 5.4 (2020): 1083-1087).

Respond to comment-6: We thank the reviewer for the thoughtful comment. As the reviewer said, the Ag/AgCl is not recommended to be used in alkaline environment for a long time test. We did the CO₂RR test within 1 hour for every applied potential, so we believe that the reported faradaic efficiency of products is reliable. The long-term test was conducted in two-electrode system without extra reference electrode. Nevertheless, we still greatly appreciate the reviewers' insightful and helpful comment

Comment 7. While detailed material characterization for Bi/VC catalyst (SEM, HRTEM) on carbon paper are reported for studies in H-cells, details on how this coating was performed on gas diffusion layers for flow cells are missing. This is especially relevant at higher current densities, since the GDE has a porous structure and reporting details on how such a stable coating was achieved are beneficial.

Respond to comment-7: We thank the reviewer for the valuable comment. As mentioned in the section of "Electrochemical measurements" in our original manuscript, for the flow-cell, the electrocatalyst inks were prepared using a mixture of 20 mg catalysts, 1.9 ml ethanol, and 0.1 ml Nafion solution. The ink was sprayed onto the GDE to yield a catalyst loading of 1 mg cm⁻². This coating method produced a mechanically stable coating, as evidenced by SEM examination of the electrode before and after the the electrochemical measurements. We collected the SEM images of BS-derived Bi and BS/VC-derived Bi/VC on gas diffusion electrode after 24 hours electrolysis in 1 M KOH at 200 mA cm⁻², and found the same phenomenon which BS-derived Bi shows a nanosheet structure and BS/VC-derived Bi/VC shows a nanowire structure. We have added the SEM image in the supplementary. We believe

these revisions according to the reviewer's comments substantially strengthen the revised manuscript.

Figure S24. SEM images of the (c) BS-derived Bi and (d) BS/VC-derived Bi/VC on gas diffusion electrode after 24 hours electrolysis in 1 M KOH at 200 mA cm^{-2} .

Comment 8. Recent studies on formate production have shown that carbonate species as reaction intermediate for CO₂RR to formate in Bi catalysts (<https://pubs.acs.org/doi/10.1021/acsomega.7b00437>). The authors should report if BS/VC catalysts create any such intermediates.

Respond to comment-8: We thank the reviewer for the valuable comment and kindness in sharing with us this interesting paper. The given paper proposed the carbonate-involved reaction mechanism by analysing the electrochemical data, but lack the in-situ characterization, such as in-situ raman and in-situ FTIR. From our in-situ ATR-SEIRAS results, we didn't find some signal change assigned to carbonate in our case. Thus, we believe the CO₂ molecule should be the reactant in the system.

Comment 9. At higher current densities, formate production is often hampered by sluggish water dissociation kinetics. Previous studies have attributed catalyst design strategies to increase formate production by increasing water dissociation kinetics. However, no discussion on this aspect is provided. It would be beneficial if discussion on how BS/VC affect water dissociation kinetics is discussed for more clarity.

Respond to comment-9: We thank the reviewer for the insightful comment. In alkaline flow-cell system, we used Ni foam as counter electrode, which is always reported in water splitting at ampere-level current density, so there is no much limitation for water dissociation in anode. As for the proton from water dissociation for CO₂ hydrogenation, we don't think the BS/VC can affect the water dissociation. According to our results, all samples we obtained show considerable formate selectivity at high current densities, indicating that the water dissociation

is not a rate-limiting factor in our case. Besides, the VC molecules have no activity toward water dissociation.

Comment 10. Some reports on In based catalysts producing formate from CO₂RR have reported that surface hydroxyl activates CO₂ reduction. (<https://pubs.acs.org/doi/10.1021/la501245p>). However, in this work the authors claim trapping of hydroxyl species increases formate production. This is contradictory and the authors needs to clarify and provide explanation for the observed differences.

Respond to comment-10: We thank the reviewer for the insightful comment. In the paper mentioned by the reviewer, the authors suggested that the formation of formate involved the insertion of CO₂ at the interface of indium and hydroxide in K₂SO₄ electrolyte. Firstly, we believe that the Bi and In have different adsorption capacity to hydroxide groups. For Bi-based catalysts, Bi sites or defective Bi sites are overwhelmingly recognized as active sites. In our theoretical calculations, the defective Bi sites have well adsorption capacity to hydroxide and it will improve the energy barrier toward *OCHO formation, and thus lower the formate selectivity. Moreover, the authors of the given paper investigate the activity of In catalysts under neutral or weakly acidic electrolyte. We conducted the CO₂RR test of sputtering In catalysts and In/VC in alkaline flow-cell with 1 M KOH. The results show that the introduction of VC layer improve the formate production at high negative potential. In view of the above trade-off effects, we think the surface hydroxide groups on indium catalysts should be controlled within a suitable coverage. We have added the relevant discussion in the revised main text. We believe these revisions substantially strengthen the revised manuscript.

Figure S38. (a) SEM image of In/VC catalyst. In electrode was synthesized by sputtering metal In target on the gas diffusion electrode using a magnetron sputtering machine. The sputtering process was carried out at a power of 1 W for 10 min. In/VC was obtained by spraying VC dispersion solution on In electrode and dried in vacuum oven for 10 hours. (b) EDS mapping of In and O elements in In/VC electrode. (c) The LSV curves of In and In/VC electrode in alkaline flow-cell system with 1 M KOH. (d) The corresponding products of In and In/VC.

Main text:

Page 21: We also investigate the feasibility of this strategy on other formate producing catalysts (**Supplementary Fig. 38-39**). The In and Sn electrode were synthesized by sputtering metal In and Sn target on the gas diffusion electrode and VC layer was sprayed on the In and Sn electrode. In previous reports, the surface hydroxyl promotes the activation of CO₂ in neutral or weakly acidic electrolytes.⁵⁵ But with VC coating, In/VC shows a lower current density but enhanced formate selectivity in alkaline electrolyte at high negative applied potential, indicating that there should be an optimal hydroxyl coverage for CO₂ activation.

Comment 11. Electrochemical active surface area for BS/VC and BS are must be calculated to show ECSA normalized current densities.

Respond to comment-11: We thank the reviewer for the constructive comment. We collected the CO₂RR performance in alkaline flow-cell system with gas diffusion electrode. The gas diffusion electrode is a porous structure carbon paper covered with carbon black and the surface area is quite large. It is hard to determine the electrochemical active surface area of catalysts and calculate the ECSA normalized current densities. But we conducted the electrochemical surface area test on Indium Tin Oxide (ITO) coated glass and investigated the ECSA change before and after CO₂RR. The results show that the active area of BS and BS/VC both increase during the CO₂RR, and the increasing trend is similar. Thus, the increased surface area is not solely responsible for the enhanced activity and selectivity on BS/VC. We have added this discussion in the revised main text. We believe these revisions substantially strengthen the revised manuscript.

Figure S27. The CV curves with different scan rate under the potential of 0.75 V – 0.85 V of BS, BS/VC, BS-after, and BS/VC-after. To avoid the affect of porous electrode (gas diffusion electrode) and electrolyte flow, we conducted the electrochemical surface area (ECSA) test by dropping the catalysts ink on Indium Tin Oxide (ITO) coated glass and run the CV in 1 M KOH without flow.

Figure S28. Δj at 0.8 V vs. RHE as a function of the scan rate to evaluate C_{dl} .

Main text:

Page 16: An electrochemical active surface area (ECSA) test was conducted on catalysts before and after electrolysis to investigate the active area change during CO₂RR (**Supplementary Fig. 27-28**). The results show that with VC coating, the BS/VC exhibited a slight decline in active area, which could be attributed to the nanowire aggregation induced by molecules. After electrolysis, the reconstruction of nanowires led to an increase of active area, whereas the active area of BS/VC-after remained lower than that of BS-after. The similar increasing trend indicates that the increase in active area is not solely responsible for the enhanced activity and selectivity on BS/VC.

Comment 12. NMR spectra and a sample calculation for quantification of formate in flow cells at high current densities must be reported.

Respond to comment-12: We thank the reviewer for the helpful comment. The NMR spectra in flow cells at high current densities is provided in Supplementary and the signal intensity only

related to the collected charge. The calculation for quantification of formate was provided in **Methods** section, and there is no difference in the quantification of formate for H-cell and flow-cell test.

Figure S22. NMR spectrum of the liquid product in alkaline flow-cell system test.

Main text:

Page 26: Liquid products were quantified by analyzing the collected electrolytes using NMR (Bruker Avance III, 600 M). A known concentration of dimethyl sulfoxide (DMSO) is used as an interior label. Based on the ¹H NMR analysis, the amounts of products were quantified by calculating the relative peak area of HCOO⁻ (1.09 ppm) and DMSO (2.6 ppm). The FE for HCOO⁻ was calculated as follows:

$$\text{FE (\%)} = \frac{nNF}{Q} \times 100\% \quad (2)$$

where n is the measured amount of HCOO⁻ (mol), N is the number of electrons required for each product, and Q is the recorded total charge during the operation.

Comment 13. At higher current densities, how much of CO₂ fed reacted with OH⁻ ions generated during the reaction? Single pass CO₂ utilization efficiency must also be reported since it's one of the important performance metrics.

Respond to comment-13: We thank the reviewer for the valuable comment. It is hard to determine how much of CO₂ fed reacted with OH⁻ ions because the remaining CO₂ is emitted directly into the air. But we can calculate the single pass CO₂ utilization efficiency at high current densities. At high negative applied potential, the BS/VC show single pass CO₂ utilization efficiency of 4.5%. We have added the relevant discussion in the revised main text. We believe these revisions substantially strengthen the revised manuscript.

Figure S23. Single pass CO₂ utilization efficiency of BS and BS/VC in alkaline flow-cell system.

Main text:

Page 13: At high negative applied potential, the BS/VC shows a single pass CO₂ utilization efficiency of 4.5%, about twice than that for BS (Supplementary Fig. 23).

Comment 14. Flooding of the GDE due to capillary pressure differences especially with KOH electrolyte is an issue. How did the authors circumvent this issue at 1.26 A/cm² reported current density?

Respond to comment-14: We thank the reviewer for the valuable comment. As the reviewer said, the flooding of the GDE in alkaline flow-cell system is a great obstacle for durability test. Thus, we conducted the durability test in all-solid electrolyte cell which can convert CO₂ to pure formic acid without going through a complex product separation and purification step, and is free from flooding issue, as also mentioned in our response to your Comment 3.

Comment 15. If poisoning by hydroxyl groups is the main hindrance for reaching higher reaction rates, can this strategy be applied for other formate producing catalysts such as Sn or In? This should be clarified and if so, preferably an experiment using Sn/VC catalyst should be performed at least in H-cells.

Respond to comment-15: We thank the reviewer for the constructive comment. As exemplified in our response to your Comment 10 above, we have conducted the CO₂RR test of sputtering In, In/VC, Sn, and Sn/VC catalysts in alkaline flow-cell with 1 M KOH. The results show that the introduction of VC layer improve the formate production of In/VC at high negative potential. Thus, we think the surface hydroxide groups on indium catalysts should be controlled within a suitable coverage. Besides, the Sn/VC shows a lower formate selectivity at a high negative applied potential, implying that the surface hydroxyl is also important for the CO₂ conversion on Sn catalysts and the VC layer can affect the distribution of hydroxyl. We have added this discussion in the revised main text. We believe these revisions following the reviewer's suggestion substantially strengthen the revised manuscript.

Figure S38. (a) SEM image of In/VC catalyst. In electrode was synthesized by sputtering metal In target on the gas diffusion electrode using a magnetron sputtering machine. The sputtering process was carried out at a power of 1 W for 10 min. In/VC was obtained by spraying VC dispersion solution on In electrode and dried in vacuum oven for 10 hours. (b) EDS mapping of In and O elements in In/VC electrode. (c) The LSV curves of In and In/VC electrode in alkaline flow-cell system with 1 M KOH. (d) The corresponding products of In and In/VC.

Figure S39. (a) SEM image of Sn/VC catalyst. Sn electrode was synthesized by sputtering metal Sn target on the gas diffusion electrode using a magnetron sputtering machine. The sputtering process was carried out at a power of 1 W for 10 min. Sn/VC was obtained by spraying VC dispersion solution on Sn electrode and dried in vacuum oven for 10 hours. (b) EDS mapping of Sn and O elements in Sn/VC electrode. (c) The LSV curves of Sn and Sn/VC electrode in alkaline flow-cell system with 1 M KOH. (d) The corresponding products of In and Sn/VC.

Main text:

Page 21: We also investigate the feasibility of this strategy on other formate producing catalysts (**Supplementary Fig. 38-39**). The In and Sn electrodes were synthesized by sputtering metal In or Sn target on the gas diffusion electrode and VC layer was sprayed on the In or Sn electrode. In previous reports, the surface hydroxyl promotes the activation of CO₂ in neutral or weakly acidic electrolytes.⁵⁵ But with VC coating, In/VC shows a lower current density but an enhanced formate selectivity in alkaline electrolyte at a high negative applied potential, indicating that there should be an optimal hydroxyl coverage for CO₂ activation. The Sn/VC

shows a similar current density with Sn, but a lower formate selectivity at a high negative applied potential, implying that the surface hydroxyl is important for the CO₂ conversion on Sn catalysts and the VC layer can affect the distribution of hydroxyl.

Comment 16. Line 76: “CORR process”. Typo in CO₂ instead of CO?

Respond to comment-16: We thank the reviewer for the careful reading and the helpful comment. We have corrected the typo in our revised manuscript.

REVIEWERS' COMMENTS

Reviewer #1 (Remarks to the Author):

The authors have effectively addressed the comments and suggestions raised by the reviewers during the initial review of the manuscript. The design principles and mechanistic insights presented in the study make a significant contribution to the CO₂ electrocatalysis community and the reviewer recommends publication of the revised manuscript.

Reviewer #2 (Remarks to the Author):

This reviewer thanks the authors for the revised version. This reviewer sees all his/her comments fully addressed.

Reviewer #3 (Remarks to the Author):

This revised manuscript at current stage shows good improvement on the study of Bi/VC catalyst design for CO₂ electrolysis to formate and the authors have addressed most questions and concerns of the reviewer. As a result, the revised manuscript can be accepted in Nature Communications journal.

However, the reviewer has some minor suggestions before final acceptance.

i) The authors responded to my comment 3 that the work's main focus was the new strategy employed of catalyst design which is indeed true and the authors have done detailed analysis and experiments which is commendable.

However, the reason for bringing this question was that this was unclear from the abstract of the manuscript. In the abstract, the authors quote "120 hours" stability, but do not mention at which current density was this stable. Could the authors make sure to state this clearly in the abstract? This is because, it might be misleading for someone reading the MS at a first glance and might expect a high performance ampere level current density for making formic acid from CO₂.

ii) In Figure S41 and other relevant figures, could the authors use 'j' instead of "I" for representing geometric current density? This is because, the symbol 'I' is widely used for representing the total current applied.

Response Letter

Manuscript ID: NCOMMS-23-05199A

Surface passivation for highly active, selective, stable, and scalable CO₂ electroreduction

Reviewer: 1

Comments to the Author

The authors have effectively addressed the comments and suggestions raised by the reviewers during the initial review of the manuscript. The design principles and mechanistic insights presented in the study make a significant contribution to the CO₂ electrocatalysis community and the reviewer recommends publication of the revised manuscript.

Response to Reviewer 1:

We thank the reviewer's approval. We strongly believe that the work described in our manuscript will have a significant positive impact on CO₂ electrocatalysis.

Reviewer: 2

Comments to the Author

This reviewer thanks the authors for the revised version. This reviewer sees all his/her comments fully addressed.

Response to Reviewer 2:

We thank the reviewer's approval. We strongly believe that the work described in our manuscript will have a significant positive impact on CO₂ electrocatalysis.

Reviewer: 3

Comments to the Author

This revised manuscript at current stage shows good improvement on the study of Bi/VC catalyst design for CO₂ electrolysis to formate and the authors have addressed most questions and concerns of the reviewer. As a result, the revised manuscript can be accepted in Nature Communications journal.

However, the reviewer has some minor suggestions before final acceptance.

Response to Reviewer 3:

We thank the reviewer's approval. We welcome the opportunity to address and clarify the issues raised in the reviewer's report and believe that the revisions substantially strengthen our manuscript. Our responses to the points raised in the report are below.

Reviewer Comments:

Comment 1. The authors responded to my comment 3 that the work's main focus was the new strategy employed of catalyst design which is indeed true and the authors have done detailed analysis and experiments which is commendable.

However, the reason for bringing this question was that this was unclear from the abstract of the manuscript. In the abstract, the authors quote "120 hours" stability, but do not mention at which current density was this stable. Could the authors make sure to state this clearly in the abstract? This is because, it might be misleading for someone reading the MS at a first glance and might expect a high performance amperic level current density for making formic acid from CO₂.

Respond to comment-1: We thank the reviewer for the valuable comment. We have added the corresponding current density in the abstract.

Abstract: When used in an all-solid-state reactor system, the newly developed BS/VC catalyst achieved efficient production of pure formic acid over 120 hours at 50 mA cm⁻² (200 mA cell current).

Comment 2. In Figure S41 and other relevant figures, could the authors use 'j' instead of "I" for representing geometric current density? This is because, the symbol 'I' is widely used for representing the total current applied.

Respond to comment-16: We thank the reviewer for the helpful comment. Sorry for the mistake. We have revised the relevant figures.

Fig. 6 | All-solid-state CO₂RR reactor performance.

Figure S41. Long-term production of pure formic acid from the BS/VC at current densities of 200 mA cm^{-2} , in the all-solid-state reactor.